# WHIMP links the actin nucleation machinery to Src-family kinase signaling during protrusion and motility

**Shail Kabrawala, Margaret D. Zimmer, Kenneth G. Campellone** [ID] *

Department of Molecular and Cell Biology, Institute for Systems Genomics, University of Connecticut, Storrs, Connecticut, United States of America

* kenneth.campellone@uconn.edu

**Data Availability Statement:** All relevant data are within the manuscript and its Supporting Information files.

## Abstract

Cell motility is governed by cooperation between the Arp2/3 complex and nucleation-promoting factors from the Wiskott-Aldrich Syndrome Protein (WASP) family, which together assemble actin filament networks to drive membrane protrusion. Here we identify WHIMP (WAVE Homology In Membrane Protrusions) as a new member of the WASP family. The *Whimp* gene is encoded on the X chromosome of a subset of mammals, including mice. Murine WHIMP promotes Arp2/3-dependent actin assembly, but is less potent than other nucleation factors. Nevertheless, WHIMP-mediated Arp2/3 activation enhances both plasma membrane ruffling and wound healing migration, whereas WHIMP depletion impairs protrusion and slows motility. WHIMP expression also increases Src-family kinase activity, and WHIMP-induced ruffles contain the additional nucleation-promoting factors WAVE1, WAVE2, and N-WASP, but not JMY or WASH. Perturbing the function of Src-family kinases, WAVE proteins, or Arp2/3 complex inhibits WHIMP-driven ruffling. These results suggest that WHIMP-associated actin assembly plays a direct role in membrane protrusion, but also results in feedback control of tyrosine kinase signaling to modulate the activation of multiple WASP-family members.

## Author summary

The actin cytoskeleton is a collection of protein polymers that assemble and disassemble within cells at specific times and locations. Sophisticated cytoskeletal regulators called nucleation-promoting factors ensure that actin polymerizes when and where it is needed, and many of these factors are members of the Wiskott-Aldrich Syndrome Protein (WASP) family. Several of the 8 known WASP-family proteins function in cell motility, but how the different factors collaborate with one another is not well understood. In this study, we identified WHIMP, a new WASP-family member that is encoded on the X chromosome of a variety of mammals. In mouse cells, WHIMP enhances cell motility by assembling actin filaments that push the plasma membrane forward. Unexpectedly, WHIMP also activates tyrosine kinases, enzymes that stimulate multiple WASP-family members during motility. Our results open new avenues of research into how nucleation factors cooperate

**Funding:** KGC was supported by National Institutes of Health grants R01-GM107441 and K02-AG050774 (www.nih.gov) and American Heart Association grant 13SDG-14640026 (www.heart.org). The funders had no role in study design, data collection and analysis, decision to publish, or preparation of the manuscript.

**Competing interests:** The authors have declared that no competing interests exist.

during movement and how the molecular activities that underlie motility differ in distinct cell types and organisms.

## Introduction

The assembly of actin filament networks is essential for many cellular functions, ranging from intracellular trafficking to whole-cell movement [1]. Branched actin polymerization is initiated by the recruitment and activation of a 7-subunit macromolecular actin nucleator named the Arp2/3 complex [2], which acts in concert with binding-partners called nucleation-promoting factors [3]. Many such factors are members of the Wiskott-Aldrich Syndrome Protein (WASP) family, and are integral in activating the complex at different cellular locations [4]. Most WASP-family proteins promote actin assembly during membrane protrusion and cell motility [5], but how the different factors collaborate during these processes is not well understood.

WASP-family members are defined by the presence of a WH2-Connector-Acidic (WCA) domain in which one or more WH2 motifs bind actin monomers, while connector and acidic peptides engage the Arp2/3 complex [6]. WCA domains induce conformational changes in the complex to promote actin nucleation and branching from the side of an existing filament [7–13]. The atypical nucleation-promoting factor Cortactin can stabilize these branches and accelerate displacement of WASP-family WCA domains to recycle them for additional Arp2/3 activation [14, 15]. In addition, the WISH/DIP1/SPIN90 family of proteins can interact with multiple nucleators and nucleation-promoting factors [16], and enables the Arp2/3 complex to create linear instead of branched filaments [17].

Eight different WASP-family proteins have been identified in mammals: WASP, N-WASP, WAVE1, WAVE2, WAVE3, WASH, WHAMM, and JMY. The first to be discovered was WASP, as mutations in the *WAS* gene give rise to X-linked immunodeficiencies [18]. WASP expression is restricted to hematopoietic cells, where it is important for development, signaling, and movement. Its closest homolog, N-WASP (Neuronal-WASP), and the more distantly related WAVEs (WASP family VErprolin homologs; also known as SCAR for Suppressor of Cyclic AMP Receptor) are expressed ubiquitously, and some are essential in mice [19–22]. These factors can be recruited to the plasma membrane and are involved in numerous endocytic or protrusive processes, including those that push the leading edge forward during cell motility [23–26]. From an evolutionary perspective, the presence of at least one WASP and one WAVE appears to be necessary for fast pseudopod-based motility [27]. Several aspects of intracellular membrane dynamics rely on other WASP-family members, including WASH (WASP and Scar Homolog), WHAMM (WASP Homolog associated with Actin, Membranes and Microtubules), and JMY (Junction Mediating regulatorY protein). WASH [28] is essential in mice [29], possibly due to its role in directing endo-lysosome trafficking [30–32]. WHAMM and JMY both drive the remodeling or transport of membranes in the secretory pathway [33–35] as well as autophagosomes [36–38]. WASH and WHAMM can also affect cell motility [39–41], likely due to their functions in membrane trafficking, while JMY can be recruited to the front of motile cells and accelerate wound healing migration [42]. Thus, all WASP-family members influence cell movement, with at least 6 playing roles at membrane protrusions.

The signaling mechanisms that promote the formation of lamellipodia and filopodia at the leading edge of moving cells can be initiated by a variety of transmembrane proteins. As examples, growth factor or adhesion receptors induce cytoskeletal responses via tyrosine kinases, lipid kinases, adaptor proteins, guanine nucleotide exchange factors (GEFs), and small G-

proteins [43–47]. Several of these molecules interact directly with WASP-family proteins. For instance, WASP and N-WASP, which are basally auto-inhibited, are activated upon binding G-proteins like Cdc42 and the phospholipid PI(4,5)P2 [26, 48–51]. Additionally, phosphorylation of specific tyrosine residues by Src- and Abl-family kinases can also stimulate WASP/ N-WASP [52–56]. The inhibition and activation of WASP/N-WASP are further influenced by interactions with actin-binding proteins from the WIP family [57]. Unlike the WASPs, the WAVEs are sequestered within multi-subunit regulatory complexes [58, 59], but engage Arp2/ 3 in response to Rac1, Arf1, and PI(3,4,5)P3 binding [60–64] and upon tyrosine phosphorylation [65–68]. WASH is also part of a regulatory complex analogous to those of the WAVEs [30, 31, 69]. It is not known if WHAMM or JMY exist in stable multi-subunit complexes, although the nucleation-promoting activity of WHAMM can be suppressed by microtubules or the small G-protein Rab1 [34, 70], and JMY-mediated actin assembly can be inhibited by the protein Strap [71].

While more than half of the WASP-family proteins can drive protrusion and motility, it is not known if mammals express additional WASP-family members that have direct roles in movement or that participate in any of the above signaling pathways. In this study, we identified and characterized a new WASP-family protein that promotes membrane protrusion and cell motility in collaboration with other nucleation factors, but with functional properties that are distinct from the previously described family members.

## Results

### WHIMP is a WASP-family protein with a WCA domain and homology to the WAVE subfamily

Given the presence of WCA domains in all WASP-family members, we searched for new mammalian nucleation factors by using the BLAST program to identify WH2 motif-containing proteins in the mouse proteome. We found an uncharacterized putative protein, Gm732, which is predicted to be 516 amino acids long and harbor a single WH2 domain and an acidic peptide near its C-terminus (Fig 1A and 1B). Moreover, a tryptophan is found as the third residue from its C-terminal end (Fig 1B). A tryptophan at this position is present in all other mammalian WASP-family members and contributes to activating the Arp2/3 complex [33, 72]. The sequence of the Gm732 WCA domain appears to be most closely related to that of WAVE1 (Fig 1C).

Bioinformatic studies indicate that WASP-family proteins are present in species separated by millions of years of evolution and also refer to *Gm732* as a putative WASP-family gene [73, 74]. Like WASP, Gm732 is encoded on the X chromosome. Using the Ensemble genome browser, we identified *Gm732* orthologs on the X in other organisms and used Geneious software to generate a translated nucleotide tree based on conserved sequence features (S1A and S1B Fig). Amino acid analyses of *Mus musculus* Gm732 suggest that it has 67% and 51% identical orthologs in rats and hamsters, respectively (S1C Fig). Although distantly-related proteins are predicted to exist in sheep, rabbits, armadillos, and elephants, no human versions of Gm732 are yet apparent.

The primary sequence of Gm732 lacks the distinct N-terminal domains that define the WASP, WAVE, WASH, or WHAMM/JMY subfamilies. Polyproline (P) peptides, which are important for binding profilin-actin, are present in every other member of the mammalian WASP family, but are also absent in Gm732. Based on the HHpred sequence similarity prediction program, Gm732 bears some resemblance to the WAVE subfamily near its N-terminus (S1D Fig). This region partially overlaps with the α6 helix within the WAVE proteins [61] and is conserved among Gm732 orthologs (S1E Fig). Given its WCA and N-terminal similarity to

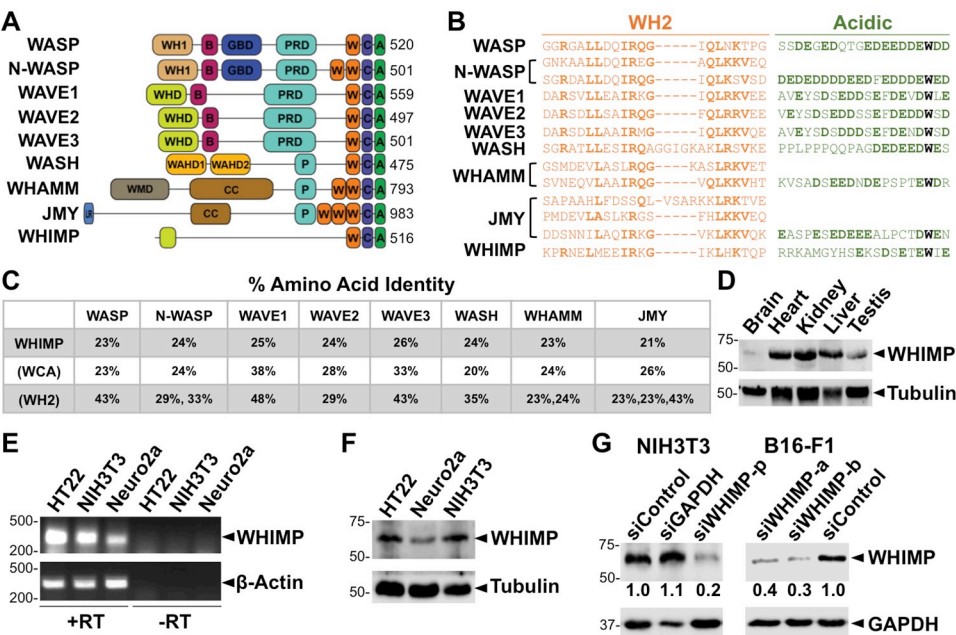

**Fig 1. WHIMP is a new WASP-family protein. (A)** The domain organizations and lengths of mouse WASP-family proteins and WHIMP (Gm732) are shown. Abbreviations: A (acidic), B (basic), C (connector), CC (coiled-coil), GBD (GTPase-binding domain), LIR (LC3-interacting region), P (polyproline), PRD (proline-rich domain), W (WASP homology 2), WAHD1-2 (WASH homology domain 1–2), WH1 (WASP homology 1), WHD (WAVE homology domain), WMD (WHAMM membrane-interaction domain). **(B)** Sequence alignments illustrate the conservation among the WH2 (orange) and Acidic (green) motifs of the WASP-family proteins. The most conserved residues in the WH2, the acidic residues in the A, and the critical tryptophan are highlighted in bold. **(C)** The % amino acid identities of full-length WHIMP, its WCA domain, and its WH2 motif were compared to other murine WASP-family proteins using EMBOSS Needle pairwise alignments. **(D)** Protein extracts were prepared from male mouse tissues, separated by SDS-PAGE (30μg/lane), and immunoblotted for WHIMP and Tubulin. **(E)** RNA isolated from HT22, NIH3T3, and Neuro2a cells was reverse transcribed (+RT) and subjected to PCR using primers for WHIMP and a β-actin control. The resulting products were visualized on ethidium bromide-stained agarose gels. PCR products were not detected when the reverse transcriptase was omitted (-RT). **(F)** HT22, Neuro2a, and NIH3T3 cell lysates were separated by SDS-PAGE and immunoblotted for WHIMP and Tubulin. **(G)** NIH3T3 and B16-F1 cells treated with either a control siRNA, a GAPDH siRNA, or WHIMP-specific siRNAs (p = pool; a,b = independent RNAs) were lysed and subjected to SDS-PAGE and immunoblotting for WHIMP, GAPDH, and Tubulin. The average quantities of WHIMP relative to Tubulin (shown beneath the WHIMP panels) were determined by densitometry of 2 representative blots.

the WAVEs, as well as our functional characterization described below, we renamed Gm732 as <u>WA</u>VE <u>H</u>omology <u>I</u>n <u>M</u>embrane <u>P</u>rotrusions–WHIMP.

While most WASP-family proteins are expressed ubiquitously within mammals, some are enriched in certain tissue types. For instance, N-WASP is expressed in most cells [75], whereas WASP is expressed in blood cells [76]. Similarly, the three WAVE proteins are expressed in many different cell types, but are elevated in neuronal cells [77]. Consequently, we examined if the WHIMP protein is expressed in mice and whether it is found in different tissues and cells. Gene Expression Atlas, an online repository of gene expression patterns, suggests that WHIMP mRNA is present in different tissue types and is more abundant in testis. To investigate this, we generated extracts derived from several mouse organs and immunoblotted them using an antibody raised against an N-terminal peptide of WHIMP. The antibody recognized a band slightly larger than WHIMP's predicted molecular weight of 55kDa in multiple tissue extracts, although the band intensities in brain and testis extracts were weaker relative to heart, kidney, and liver (Fig 1D). Based on these results, it appears that the WHIMP protein is broadly expressed in murine tissues.

We next examined WHIMP expression in several mouse cell lines via RT-PCR and by immunoblotting. RT-PCR using RNA extracted from HT22 and Neuro2a neuronal cell lines, as well as NIH3T3 fibroblasts, demonstrated that WHIMP was transcribed in these cells (Fig 1E). Using protein extracts from the same cell lines, we immunoblotted for WHIMP and found that WHIMP protein was expressed as well (Fig 1F). Additionally, to confirm the specificity of the WHIMP antibody, we treated NIH3T3 fibroblasts with a pool of siRNAs targeting WHIMP and treated another cell line, B16-F1 melanoma cells, with independent siRNAs to WHIMP. Immunoblotting revealed that the intensity of the ~60kDa band was typically reduced by 80% in NIH3T3 cells and by about 70% in B16-F1 cells (Fig 1G). These results indicate that WHIMP is expressed in multiple mouse tissues and cell lines.

## WHIMP is a weak actin nucleation-promoting factor *in vitro* and in cells

Since WHIMP is predicted to have a WCA domain, we sought to test whether it could promote actin nucleation *in vitro*. We therefore cloned the WHIMP cDNA sequence into a plasmid encoding an N-terminal maltose binding protein (MBP) tag, then expressed and isolated recombinant MBP-WHIMP from baculovirus-infected insect cells. MBP-WHIMP was poorly soluble and prone to proteolysis, as multiple WHIMP degradation products were detectable by Coomassie blue staining and immunoblotting for the MBP tag or N-terminus of WHIMP (S2A Fig). Nevertheless, pyrene-actin assembly assays predictably demonstrated that, in the presence of the Arp2/3 complex, MBP-WHIMP caused a concentration-dependent increase in actin polymerization relative to an MBP control (S2B Fig). This was reflected in increases in maximum assembly rates (S2C Fig) and decreases in times to reach half-maximum polymer (S2D Fig). In the absence of Arp2/3, no significant WHIMP-dependent changes in actin assembly were detected (S2B Fig). In order to gauge WHIMP's potency, we compared its activity to MBP-tagged WHAMM, which was previously shown to be a relatively weak nucleation-promoting factor [33]. These experiments suggested that WHIMP was a less potent activator of the Arp2/3 complex than WHAMM (S2B–S5D Figs).

WCA domains are sufficient to initiate actin nucleation by binding G-actin and the Arp2/3 complex, so to test if the WHIMP WCA domain could induce actin assembly in an Arp2/3-dependent or independent manner, we purified an MBP-WHIMP(WCA) fusion protein (Fig 2A). Previous *in vitro* studies demonstrated that JMY(WWWCA) can promote Arp2/3-independent actin nucleation [42], and as a positive control in our experiments, JMY(WWWCA) caused a significant increase in the actin assembly rate in the absence of the Arp2/3 complex (Fig 2B). However, no such increase was observed with WHIMP(WCA). Only in the presence of the Arp2/3 complex did WHIMP(WCA) accelerate actin polymerization (Fig 2B).

To gain a more quantitative assessment of the potency with which WHIMP increases Arp2/3-mediated actin polymerization, we next compared actin assembly rates driven by WHIMP(WCA) and by WCA domains from 6 other members of the WASP family (Fig 2A) at a range of concentrations. In the presence of 20nM Arp2/3 complex and WCA concentrations of 50nM (Fig 2C) and 200nM (Fig 2D), WASP and WAVE2 stimulated the fastest actin polymerization kinetics, whereas WHIMP stimulated the slowest. Furthermore, quantification of polymerization curves indicated that the maximum actin assembly rates for 500nM WASP(WCA) and WHIMP(WCA) were 46nM/s and 3nM/s, respectively (Fig 2E). Although actin polymerization mediated by WHIMP(WCA) was slow even at this high concentration, it was still 3-fold faster than the MBP control. Consistent with these results, calculations of the times to half maximum polymer for 500nM WASP(WCA), WHIMP(WCA), and MBP were 29s, 355s, and 700s, respectively (Fig 2F). Overall, the WCA domains could be placed into a hierarchy based on their potency, where WASP, WAVE2 > N-WASP, JMY > WHAMM > WASH > WHIMP.

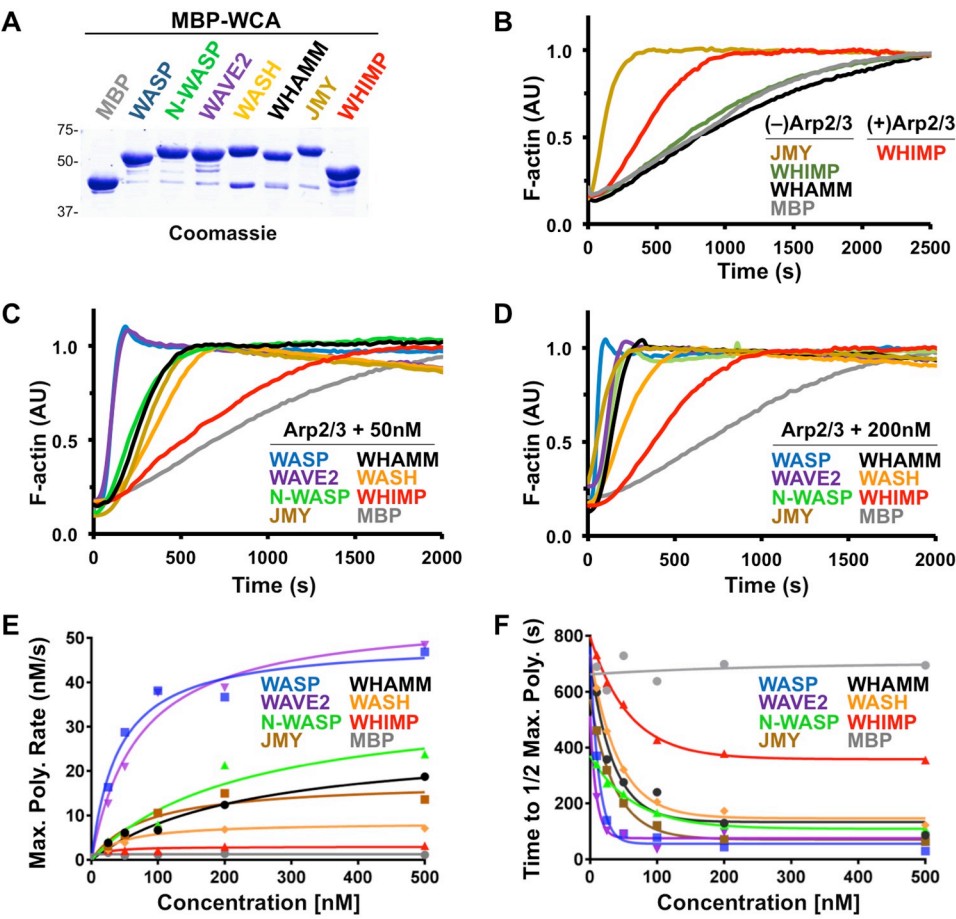

**Fig 2. WHIMP is a weak Arp2/3-dependent actin nucleation factor *in vitro*.** (A) Purified maltose binding protein (MBP) tagged WCA domains from WASP-family proteins (5μg/lane) were subjected to SDS-PAGE followed by staining with Coomassie blue. (B) Representative actin polymerization assays using 2μM actin, with or without 20nM Arp2/3 complex, and 200nM MBP-JMY(WWWCA), MBP-WHIMP(WCA), MBP-WHAMM(WWCA), or MBP are shown. AU, arbitrary units. (C-D) Actin polymerization assays were performed using 20nM Arp2/3 complex in the presence of 50nM or 200nM MBP-WCA fusion proteins. (E) Maximum actin polymerization rates driven by the Arp2/3 complex plus different concentrations of MBP-WCA fusion proteins were calculated. (F) Times to half-maximal polymer formed by the Arp2/3 complex plus different concentrations of MBP-WCA fusion proteins were calculated.

To test whether WHIMP might enhance or inhibit the nucleation-promoting activity of another WASP-family protein, we combined the MBP-tagged WCA domains of WAVE2 and WHIMP in pyrene-actin assembly assays. In the presence of the Arp2/3 complex and a low concentration of WAVE2(WCA), an expected increase in actin polymerization was observed (S2E Fig). When Arp2/3 and WAVE2(WCA) were mixed with an excess of either MBP or MBP-WHIMP(WCA), no significant changes to actin assembly rates were detected (S2E Fig), suggesting that WHIMP(WCA) does not have an obvious synergistic or inhibitory effect on actin assembly in the presence of another WCA domain.

Given the weak nucleation-promoting activity of WHIMP in pyrene-actin polymerization assays, we wondered whether this lack of potency was intrinsic to WHIMP or if it could be due to an absence of necessary cofactors under the *in vitro* conditions. To test this, we examined whether overexpression of full-length WHIMP or the WHIMP WCA domain could promote actin assembly in cells by transiently transfecting NIH3T3 fibroblasts with plasmids encoding

GFP-WHIMP, GFP-WHIMP(WCA), or GFP-N-WASP(WWCA) as a control (Fig 3A). All of the GFP-tagged proteins were found predominantly in the cytosol, but full-length GFP-WHIMP localized to the cell periphery whenever cells contained membrane protrusions or ruffles (Fig 3B; see below). Quantification of phalloidin-stained F-actin in cells expressing GFP-WHIMP(WCA) or GFP-WHIMP revealed an average increase in F-actin content of 20–25% relative to cells expressing GFP alone (Fig 3C). By comparison, cells expressing GFP-N-WASP(WWCA) showed a 2-fold increase in F-actin fluorescence (Fig 3C). Transient expression of GFP-WHIMP, GFP-WHIMP(WCA), or GFP-N-WASP(WWCA) in B16-F1 cells also resulted in higher levels of F-actin staining compared to untransfected cells or to cells expressing GFP (S3A and S3B Fig). Thus, our biochemical and cellular experiments both demonstrate that WHIMP is an actin nucleation-promoting factor, albeit a relatively weak one.

## WHIMP can be recruited to membrane protrusions

To begin to characterize where WHIMP is found within cells, we more closely examined tagged WHIMP localization following transient transfections. GFP-WHIMP was present predominantly in the cytoplasm, but in a fraction of NIH3T3 cells (Fig 3B) and even moreso in B16-F1 cells (S3B Fig), it was enriched in portions of the cell periphery. To explore the possibility that WHIMP might be recruited to the plasma membrane without multi-plasmid based overexpression, and under distinct culture conditions, we generated NIH3T3 cell lines stably encoding a LAP (Localization and Affinity Purification) tag [78], LAP-WHIMP, mCherry, or mCherry-WHIMP (S4A Fig). Inducible expression of the LAP- and mCherry-fusion proteins was confirmed by immunoblotting and fluorescence microscopy (S4B and S4C Fig). LAP-WHIMP was typically expressed at 4-fold higher levels than endogenous WHIMP (S4D Fig), and treatment with siRNAs to WHIMP resulted in a nearly-complete reduction in LAP-WHIMP and a partial depletion of endogenous WHIMP (S4E and S4F Fig).

To more thoroughly assess WHIMP recruitment to the cell periphery, we next observed cells shortly after seeding onto coverslips, during cell spreading. Notably, mCherry-WHIMP, but not mCherry, co-localized extensively with F-actin at peripheral membrane protrusions (Fig 3D). We also examined WHIMP localization when serum-starved cells were treated with epidermal growth factor (EGF) to increase membrane ruffling. Consistent with the spreading assays, mCherry-WHIMP became enriched at membrane protrusions within minutes after EGF stimulation (Fig 3E). A similar redistribution of LAP-WHIMP also took place after EGF treatment (S4G Fig). Line-scan analyses of mCherry-WHIMP and F-actin fluorescence intensities further demonstrated that WHIMP localization to F-actin-rich protrusions increased in response to EGF (Fig 3F). Collectively, these results indicate that WHIMP can be recruited to the plasma membrane under multiple culture conditions.

## WHIMP-associated membrane ruffles contain the Arp2/3 complex, WAVEs, and N-WASP

To ascertain whether WHIMP-associated membrane protrusions contain the Arp2/3 complex, we stained the cell lines that express LAP-WHIMP or LAP with antibodies to the Arp3 and ArpC2 subunits of the complex. As expected, both Arp3 and ArpC2 were consistently found in the F-actin-rich extensions of LAP-WHIMP-expressing cells (Fig 4A). Moreover, line-scan analyses revealed overlaps in intensities between the Arp subunit staining and LAP-WHIMP, but not LAP, at the protrusions (Fig 4B and 4C; S5A and S5B Fig). 3D-confocal imaging confirmed the coinciding localization patterns of Arp3 and LAP-WHIMP at ruffle-like structures, and orthogonal views showed Arp3 and LAP-WHIMP along the height of dorsal protrusions (Fig 4D).

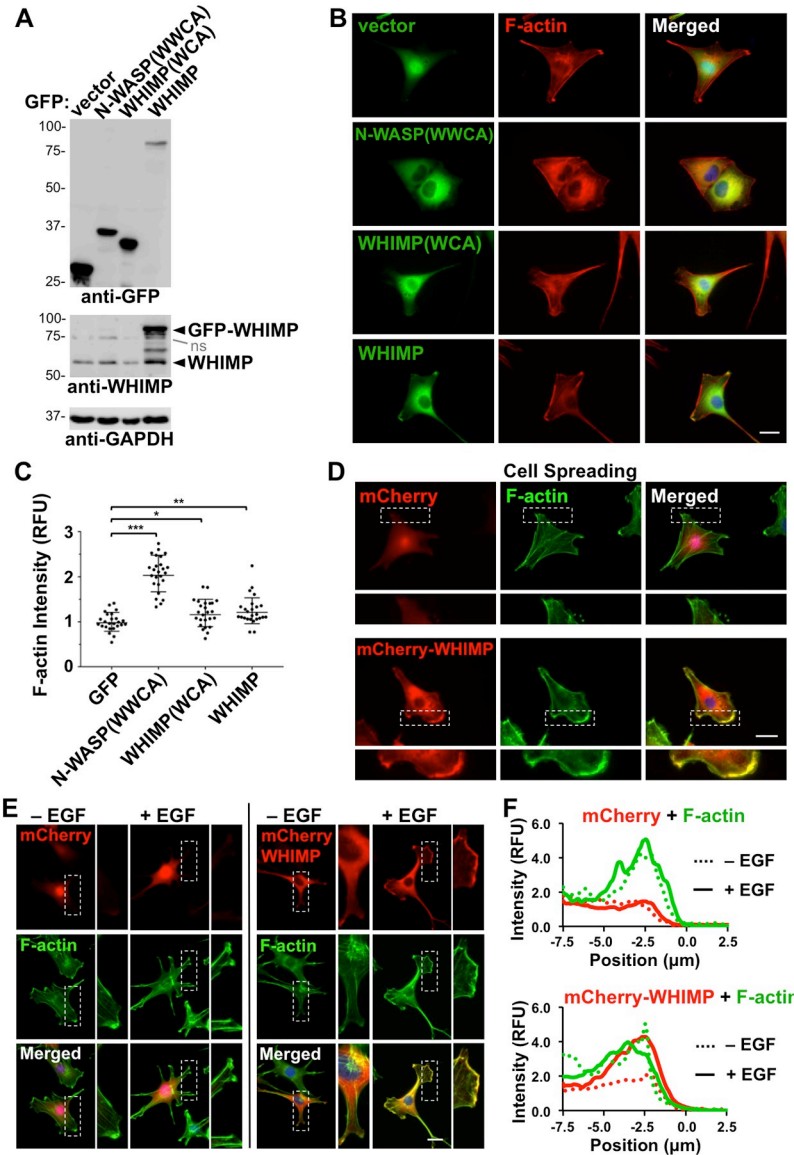

Fig 3. WHIMP promotes actin assembly in cells and can be recruited to membrane protrusions. (A) NIH3T3 cells transiently transfected with plasmids encoding GFP or GFP-tagged N-WASP(WWCA), WHIMP(WCA), or full-length WHIMP were lysed and subjected to SDS-PAGE and immunoblotting with antibodies to GFP, WHIMP, and GAPDH. ns, non-specific. (B) Cells treated as in A were fixed and stained with phalloidin to visualize F-actin (red) and DAPI to label DNA (blue). (C) Mean F-actin fluorescence intensities were measured for cells expressing the GFP-fusion proteins and normalized to F-actin intensity in cells expressing GFP alone. Each point represents the intensity of a single cell (n = 25 total per construct pooled from 3 experiments), and the horizontal line represents their mean +/-SD. (D) NIH3T3 cells stably encoding mCherry or mCherry-WHIMP were induced to express the fusion proteins, seeded onto coverslips, and allowed to adhere and spread for 3h. Cells were then fixed and stained with phalloidin to visualize F-actin (green) and DAPI to label DNA (blue). (E) NIH3T3 cells stably encoding mCherry or mCherry-WHIMP were induced, serum-starved, and stimulated with EGF for 5min. Cells were then fixed and stained with phalloidin and DAPI. Magnifications in D-E highlight mCherry-WHIMP-specific enrichment at the cell periphery and co-localization with F-actin. (F) Line-scan plots depict the mean pixel intensity of mCherry, mCherry-WHIMP, and phalloidin fluorescence near the edges of cells (n = 3). F-actin intensity was normalized to the signal beneath the nucleus, and the edge of the cell was set to 0µm on the X-axis. RFU, relative fluorescent units. *** p<0.001, ** p<0.01, * p<0.05 (ANOVA with Tukey post-test). Scale bars, 20µm.

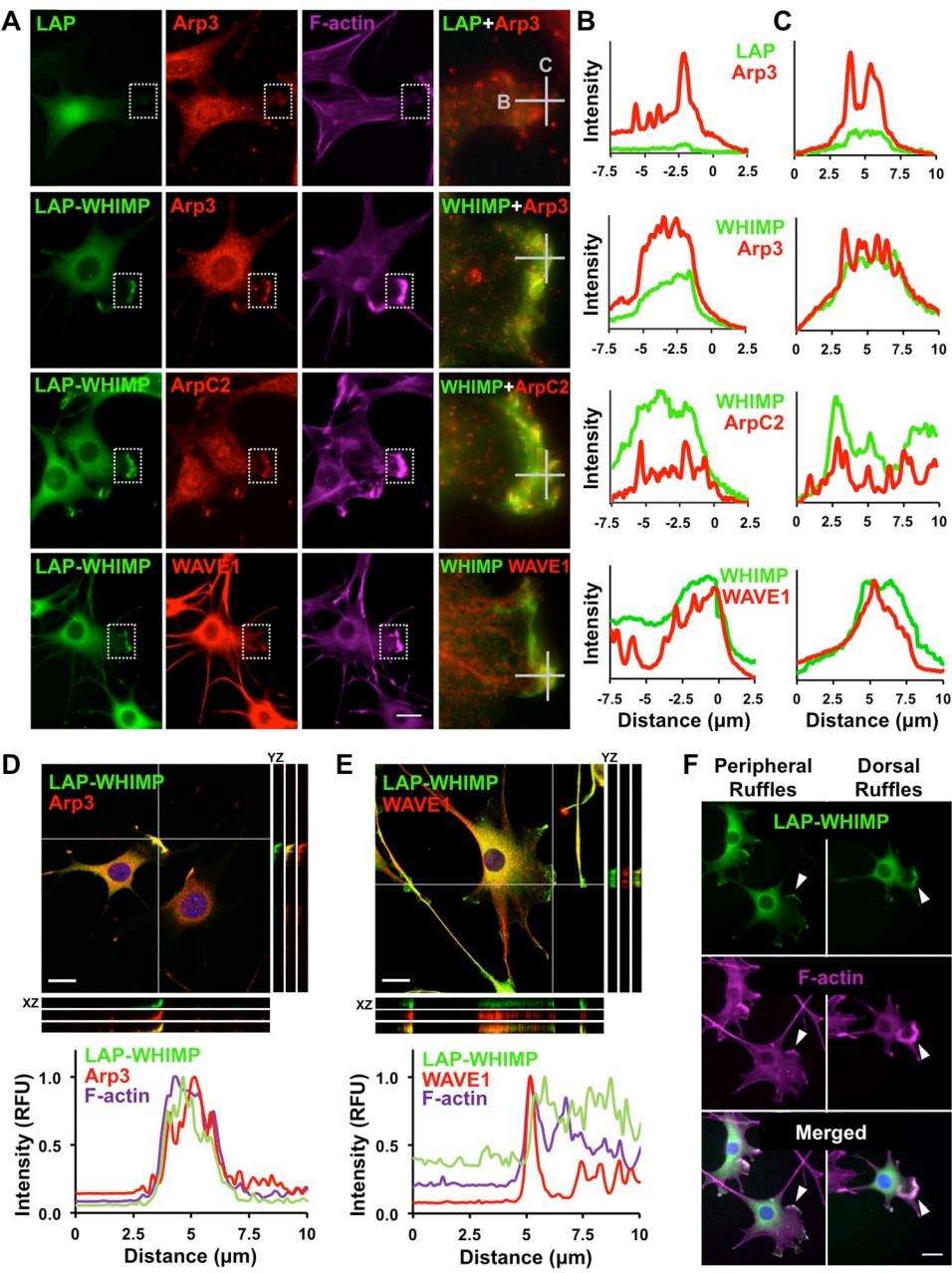

**Fig 4. WHIMP-associated membrane protrusions contain the Arp2/3 complex and WAVE1. (A)** NIH3T3 cell lines stably encoding a Localization and Affinity Purification (LAP) tag or LAP-WHIMP were induced to express the fusion proteins, fixed, and stained with phalloidin to visualize F-actin (magenta), antibodies to Arp3, ArpC2, or WAVE1 (red), and DAPI to label DNA (blue). Magnifications (right column) highlight LAP-WHIMP-specific localization to plasma membrane protrusions and overlap with the Arp2/3 complex and WAVE1. **(B-C)** Line-scan plots depict the mean pixel intensity of LAP, LAP-WHIMP, Arp3, ArpC2, and WAVE1 along the horizontal and vertical 10μm lines near the edges of the cells. The edges were set to 0μm on the X-axes. **(D-E)** Cells stained as in A were subjected to confocal microscopy. Maximum intensity projections are shown in the large panels, and adjacent orthogonal views of YZ and XZ planes are indicated by the two gray lines. The plot profiles depict pixel intensities of LAP-WHIMP, Arp3, WAVE1, and F-actin along a 10μm line drawn near the intersection of the two gray lines. **(F)** Cells were treated as in A, and examples of WHIMP-associated peripheral and dorsal ruffles are shown. Scale bars, 20μm.

To determine if other WASP-family members were also present in the protrusions, we stained LAP-WHIMP-expressing cells with antibodies to WAVE1, WAVE2, N-WASP, JMY, and WASH. Among these proteins, JMY and WASH were absent from LAP-WHIMP-associated membrane ruffles, while WAVE1, WAVE2, and N-WASP were present (Fig 4A; S5A Fig). The absence of JMY and WASH aligns with previous work indicating that JMY does not play an important role in mouse fibroblast protrusion [79] and that WASH resides on endomembranes [30–32]. The observations that the WAVEs and N-WASP were regularly found in the WHIMP-containing protrusions could be expected based on numerous studies of their localizations, and is consistent with the idea that multiple nucleation factors activate Arp2/3 in these structures.

Since WHIMP appears to be most related to WAVE1, we examined the localization of WAVE1 more thoroughly. Line-scan analyses revealed overlap between WAVE1 and LAP-WHIMP within ruffles, although some areas contained more LAP-WHIMP than WAVE1, while in other areas the distribution was reversed (Fig 4B, 4C and 4E). Collectively, these results suggest that during WHIMP-associated plasma membrane remodeling, WHIMP localization parallels that of the Arp2/3 complex, but also involves the recruitment of N-WASP and the WAVE subfamily of nucleation-promoting factors.

## WHIMP expression induces the assembly of large, fast-moving membrane protrusions

During our examinations of actin assembly factor localization, it was evident that F-actin-rich membrane ruffles were much more prominent in the LAP-WHIMP-expressing cell line than in cells expressing the LAP control protein. Such ruffles were noticeable in both peripheral and dorsal locations (Fig 4F). To quantify ruffling in LAP- versus LAP-WHIMP-expressing cell lines, and to assess the role of the Arp2/3 complex in generating these protrusions, we next measured the fraction of cells with ruffles when cultured in the absence or presence of the Arp2/3 inhibitor CK666. Under control DMSO-treated conditions, nearly 70% of LAP-WHIMP-expressing cells possessed F-actin-rich ruffles, compared to only 10% of LAP-expressing cells (Fig 5A and 5B). NIH3T3 cell lines expressing GFP-N-WASP or GFP-Cortactin did not exhibit such pronounced ruffling phenotypes (S6A Fig), indicating that a ruffle-inducing effect was specific to WHIMP and not a generic consequence of overexpressing a nucleation-promoting factor.

Next, we added CK666 to cells for 1h to inhibit the Arp2/3 complex prior to microscopic examination. Compared to DMSO-treated controls, ruffling was significantly reduced by CK666, as less than 30% of LAP-WHIMP-expressing cells contained ruffles, while LAP-expressing cells were devoid of protrusions (Fig 5A and 5B). To confirm the importance of Arp2/3 in ruffling dynamics, we examined live LAP-WHIMP-expressing cells before and during treatment with CK666. Timelapse imaging revealed that at steady-state the untreated cells were highly dynamic, as WHIMP-associated membranes ruffled consistently (Fig 5C). In contrast, the addition of CK666 caused ruffling activity to stop within minutes, whereas removal of CK666 resulted in a burst of protrusion and resumption of ruffling (Fig 5C). Together, these experiments indicate that elevating the expression level of WHIMP, but not distantly-related WASP-family members, is sufficient to induce the assembly of large F-actin-rich membrane ruffles, and that the protrusion of WHIMP-associated structures requires Arp2/3 complex activity.

Previous studies of all WASP-family proteins have shown that the WCA domain is essential for activating the Arp2/3 complex, as deletion of this domain results in a loss of nucleation-promoting activity [26, 33, 42, 72]. Consequently, to test whether WHIMP-associated membrane ruffles were formed in a WCA-dependent manner, we generated NIH3T3 cells stably

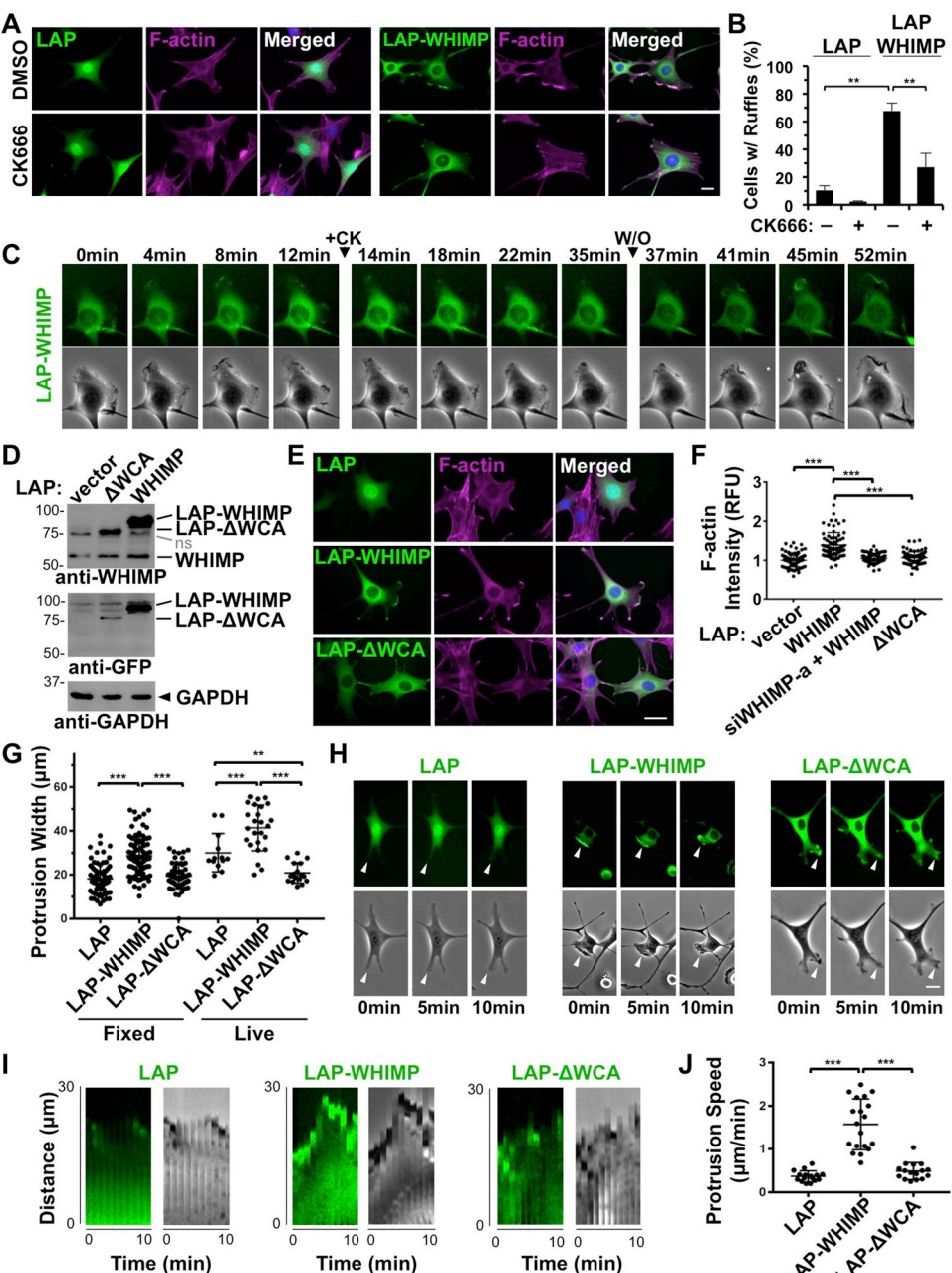

**Fig 5. WHIMP expression induces the assembly of large, fast-moving membrane protrusions. (A)** NIH3T3 cell lines expressing LAP or LAP-WHIMP were treated with DMSO or 100μm CK666 for 1h, fixed, and stained with phalloidin to visualize F-actin (magenta) and DAPI to label DNA (blue). **(B)** The % of cells with ruffles was quantified from experiments performed in A. Each bar represents the mean % of cells (n = 150–200 per condition) with ruffles from 2 such experiments +/- SD. **(C)** GFP fluorescence and phase-contrast images of live LAP-WHIMP-expressing cells were captured every minute. The times of addition (+CK) and washout (W/O) of 200μM CK666 are shown. The pattern of ruffling, reduced ruffling, and resumed ruffling were observed in n>20 cells across 3 experiments. **(D)** Cells expressing LAP, LAP-WHIMP, or LAP-WHIMP(ΔWCA) were subjected to SDS-PAGE and immunoblotting with antibodies to WHIMP, GFP, and GAPDH. ns, non-specific. **(E)** Cells expressing LAP, LAP-WHIMP, or LAP-WHIMP (ΔWCA) were fixed and stained with phalloidin and DAPI. **(F)** Mean F-actin fluorescence intensities were measured for cells encoding the LAP-fusion proteins and normalized to F-actin intensity in cells expressing LAP alone. Each point represents the intensity of a single cell (n>70 per construct pooled from 2–3 experiments), and the horizontal line represents their mean +/-SD. **(G)** Membrane protrusion widths were measured in fixed (n>60) and live (n>14) cells expressing different LAP-tagged constructs in 2–3 experiments. Protrusive regions were identified by examining F-actin enrichment at cell edges in fixed cells, and phase-contrast images in live cells. Each point represents a single

cell, and the horizontal line represents their mean +/-SD. **(H)** GFP fluorescence and phase-contrast timelapse images of live LAP-, LAP-WHIMP-, or LAP-WHIMP(ΔWCA)-expressing cells were captured as in panel C. Arrowheads highlight examples of ruffles. **(I)** Membrane protrusions observed during live imaging are shown in the representative kymographs. **(J)** Maximum protrusion speeds were calculated from live samples in G based on the distance achieved by the largest continuous protrusion in the smallest amount of time within a 10min window of imaging. Each point represents protrusion speed from a single cell (n>15), and horizontal lines represent their mean +/-SD. *** $p < 0.001$, ** $p < 0.01$ (ANOVA with Tukey post-test). Scale bars, 20μm.

encoding a LAP-WHIMP mutant lacking the WCA domain (ΔWCA) for comparisons to LAP and LAP-WHIMP cells. Expression of LAP-WHIMP(ΔWCA) was verified via immunoblotting and fluorescence microscopy (Fig 5D and 5E), although a smaller percentage of cells possessed visibly detectable levels of this mutant (see Methods). As predicted, compared to full-length LAP-WHIMP, LAP-ΔWCA failed to increase cytoplasmic actin assembly, as the F-actin levels in ΔWCA-expressing cells matched those found in LAP vector control or WHIMP siRNA-treated cells (Fig 5E and 5F).

Upon examination of LAP-WHIMP- and LAP-ΔWCA-expressing cells, it was noticeable that cells expressing the WHIMP mutant had less prominent ruffles (Fig 5E). To quantify the impact of the WCA deletion on ruffling, we measured protrusion widths in both fixed and live cells. In fixed conditions, protrusions in LAP-WHIMP-expressing cells were approximately 50% wider than those in LAP control cells, and this increase in size was abrogated by the ΔWCA mutation (Fig 5G). A similar pattern was observed in live cells, except that cells expressing the WHIMPΔWCA mutant had significantly smaller ruffles than those found in LAP control cells (Fig 5G), suggesting that this construct is capable of inhibiting ruffling. Moreover, timelapse imaging revealed that the plasma membranes of LAP-WHIMP-expressing cells were much more active than those of LAP- or LAP-ΔWCA-expressing cells (Fig 5H). To quantify the effect of the WCA deletion on ruffling dynamics, we measured protrusion speeds in live cells. Kymographs confirmed that expression of full-length WHIMP was associated with greater protrusive activity (Fig 5I; S4H Fig) and demonstrated that protrusion speeds in LAP-WHIMP cells were almost three times what were observed in LAP- or LAP-ΔWCA-expressing cells (Fig 5J). Collectively, these studies support the conclusion that WHIMP WCA-dependent activation of the Arp2/3 complex promotes the formation of large, fast-moving membrane protrusions.

## WHIMP enhances macropinocytosis

When imaging ruffling in live cells, we often observed the formation of circular structures resembling macropinocytic cups at the dorsal surfaces of cells expressing LAP-WHIMP (Fig 6A). Previous studies have shown that the WAVE proteins are important for generating similar ruffles, and that a deletion of WAVE1 inhibits the formation of dorsal protrusions [80, 81]. To test whether WHIMP enhances macropinocytosis, we incubated cells expressing LAP-WHIMP with a fluorescent dextran that is subjected to fluid-phase receptor-independent internalization. Following a 60 minute incubation, LAP-WHIMP cells contained significantly more fluorescent dextran than LAP control cells (Fig 6B and 6C), suggesting that WHIMP expression increases macropinocytosis. As further controls for evaluating whether WHIMP participates in other forms of internalization, we incubated GFP-WHIMP-expressing cells with fluorescent transferrin, which is taken up by receptor-mediated endocytosis, and also looked for co-localization between GFP-WHIMP and the early endosome marker Rab5. GFP-WHIMP did not substantially co-localize with transferrin (Fig 6D) or with mCherry-Rab5 (Fig 6E). These results are consistent with WHIMP being important for the protrusion-

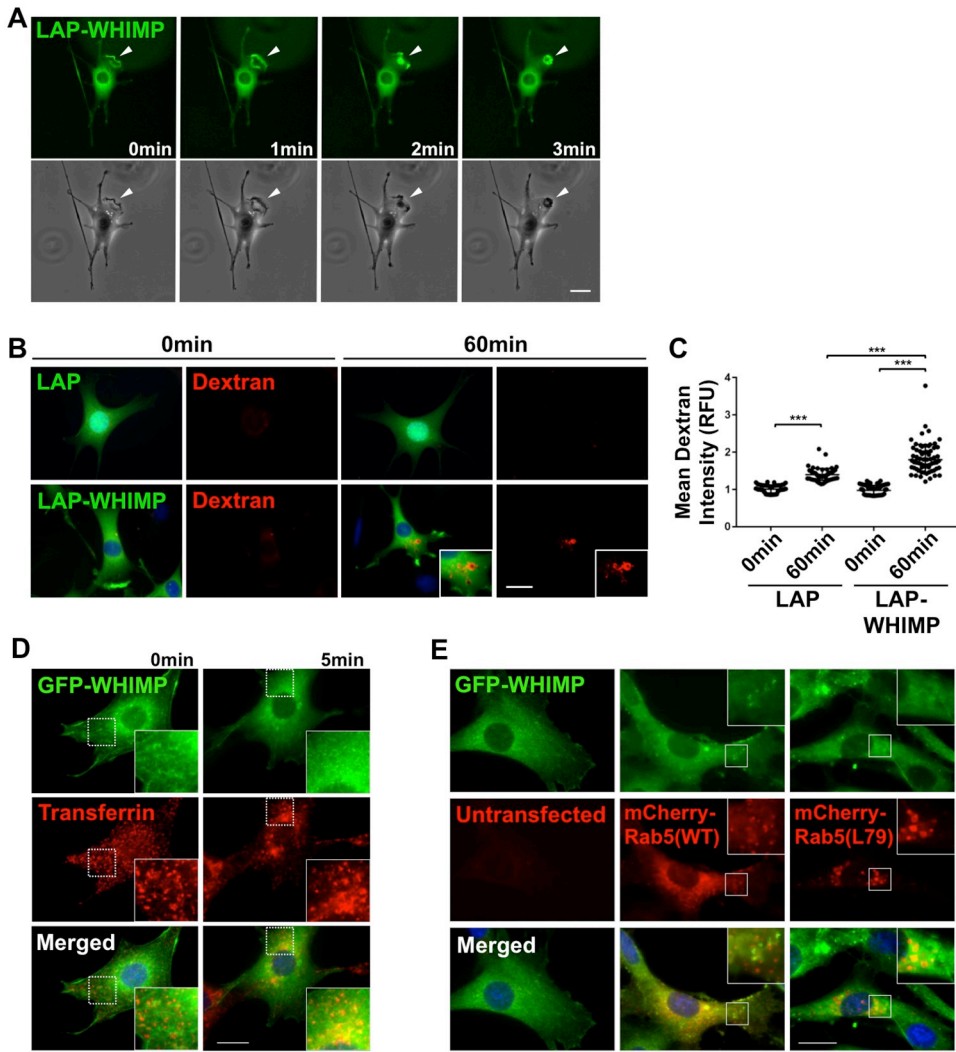

**Fig 6. WHIMP expression enhances macropinocytosis. (A)** GFP fluorescence and phase-contrast timelapse images of live LAP-WHIMP-expressing NIH3T3 cells are shown. Arrowheads highlight dorsal ruffle formation and circular constriction. **(B)** Cells expressing LAP or LAP-WHIMP were incubated with 70kDa TMR-dextran (red) for 0 or 60min, fixed, and stained with DAPI to label DNA (blue). Insets highlight increased amounts of TMR-dextran in LAP-WHIMP cells after 60min. **(C)** The mean TMR-dextran intensity per cell was measured using ImageJ. Each point represents the mean intensity of a single cell (n≥60) compiled from multiple experiments, and the horizontal line represents the overall mean +/-SD. **(D)** NIH3T3 cells were transiently transfected with a plasmid encoding GFP-WHIMP, incubated with Alexa568-transferrin at 4˚C, then shifted to 37˚C for 5min, fixed, and stained with DAPI. Insets highlight a lack of co-localization of GFP-WHIMP with punctate transferrin. **(E)** Cells were transiently co-transfected with plasmids encoding GFP-WHIMP and either wild type Rab5a or constitutively active Rab5a(Q79L), fixed, and stained with DAPI. Insets highlight a lack of co-localization between GFP-WHIMP and either Rab5a or large endosome-associated Rab5a(Q79L). *** p<0.001 (ANOVA with Tukey post-test). Scale bars, 20μm.

mediated process of non-selective macropinocytosis rather than receptor-mediated forms of endocytosis.

## WHIMP expression positively correlates with wound closure rates

It is well established that membrane protrusion is a key step in cell motility, and studies on the WAVE proteins, N-WASP, and JMY have shown that multiple nucleation factors can be

recruited to the plasma membrane to drive protrusion [5, 24, 25, 42, 82]. Given the localization of WHIMP in our experiments, we next examined its function in cell motility. NIH3T3 cell lines expressing LAP or LAP-WHIMP were each treated with control or WHIMP siRNAs and seeded onto glass-bottom dishes containing silicone inserts for separating groups of cells. Immunoblotting of lysates from siWHIMP-treated LAP-expressing cells typically revealed an approximate 80% depletion of endogenous WHIMP (Fig 7A). The same treatment in cells expressing LAP-WHIMP resulted in a 70% depletion of endogenous WHIMP and a 90% depletion of tagged-WHIMP (Fig 7A). When the inserts were removed, 0.5mm wide 'wound' areas remained (Fig 7B). In live cells at the edge of the wound area, LAP-WHIMP localized to protrusions, but upon siWHIMP treatment this signal was lost in most cells (Fig 7C). These results indicate that WHIMP can be recruited to the leading edge of migrating cells and that this localization can be abrogated following siRNA treatment. We also fixed LAP- and LAP-WHIMP-expressing cells and stained them for actin filaments. As expected, WHIMP localization overlapped with intense F-actin staining at the front of cells (Fig 7D). These findings are consistent with WHIMP having a function at the leading edge of motile cells.

To evaluate the efficiency with which wounds fill under the four different experimental conditions (Fig 7A and 7B), we captured images of wound areas at several intervals and measured the remaining unoccupied areas. First, for examining WHIMP loss-of-function, we compared LAP-expressing cells treated with siControl versus siWHIMP RNAs. In these experiments, the siWHIMP-treated cells exhibited less efficient wound closure at 12h and 24h, with an unfilled wound area that was approximately 4-fold greater than that of siControl-treated LAP cells at the latter timepoint (Fig 7E), demonstrating that depletion of endogenous WHIMP results in slower wound healing. Second, we compared siControl-treated cells expressing LAP with those expressing LAP-WHIMP and found that the WHIMP-overexpressing cells had significantly smaller wound areas at 6h, 12h, and 24h (Fig 7E), suggesting that increasing WHIMP protein levels accelerates wound healing. Finally, to determine if this LAP-WHIMP-associated enhancement in wound closure could be counteracted, we compared the behaviors of LAP-WHIMP cells following siControl versus siWHIMP treatments. At 6h, the remaining wound area for siWHIMP-treated LAP-WHIMP cells was slightly larger than the empty area for siControl cells, and by 12h the difference was greater, with siWHIMP cells covering 20% less of the wound than siControl cells (Fig 7E). For both timepoints, the samples in which WHIMP overexpression was recalibrated using WHIMP siRNAs displayed wound areas that more closely resembled the areas in LAP control samples. By 24h, the wound areas for both the siControl and siWHIMP-treated LAP-WHIMP-encoding cells were almost completely covered (Fig 7E), although fewer siWHIMP-treated cells were present in the wound area (Fig 7B and 7F). Overall, these assays indicate that WHIMP depletion in fibroblasts inhibits wound healing, that LAP-WHIMP overexpression enhances the closure process, and that re-adjusting total WHIMP expression to near-normal levels can restore wound filling to near-normal rates.

## WHIMP and the WAVE proteins both contribute to rapid cell motility and efficient protrusion

To further characterize the function of WHIMP in migration, we studied the effects of depleting WHIMP from the B16-F1 mouse melanoma cell line, which is considered a highly motile epithelial cell type and generates more pronounced protrusions than NIH3T3 fibroblasts. The B16-F1 cells were treated with WHIMP siRNAs and seeded in glass bottom dishes containing silicone inserts, as described above (Fig 8A and 8B). Immunoblotting typically revealed a 70–80% depletion of WHIMP in samples receiving independent WHIMP siRNAs (Fig 8A).

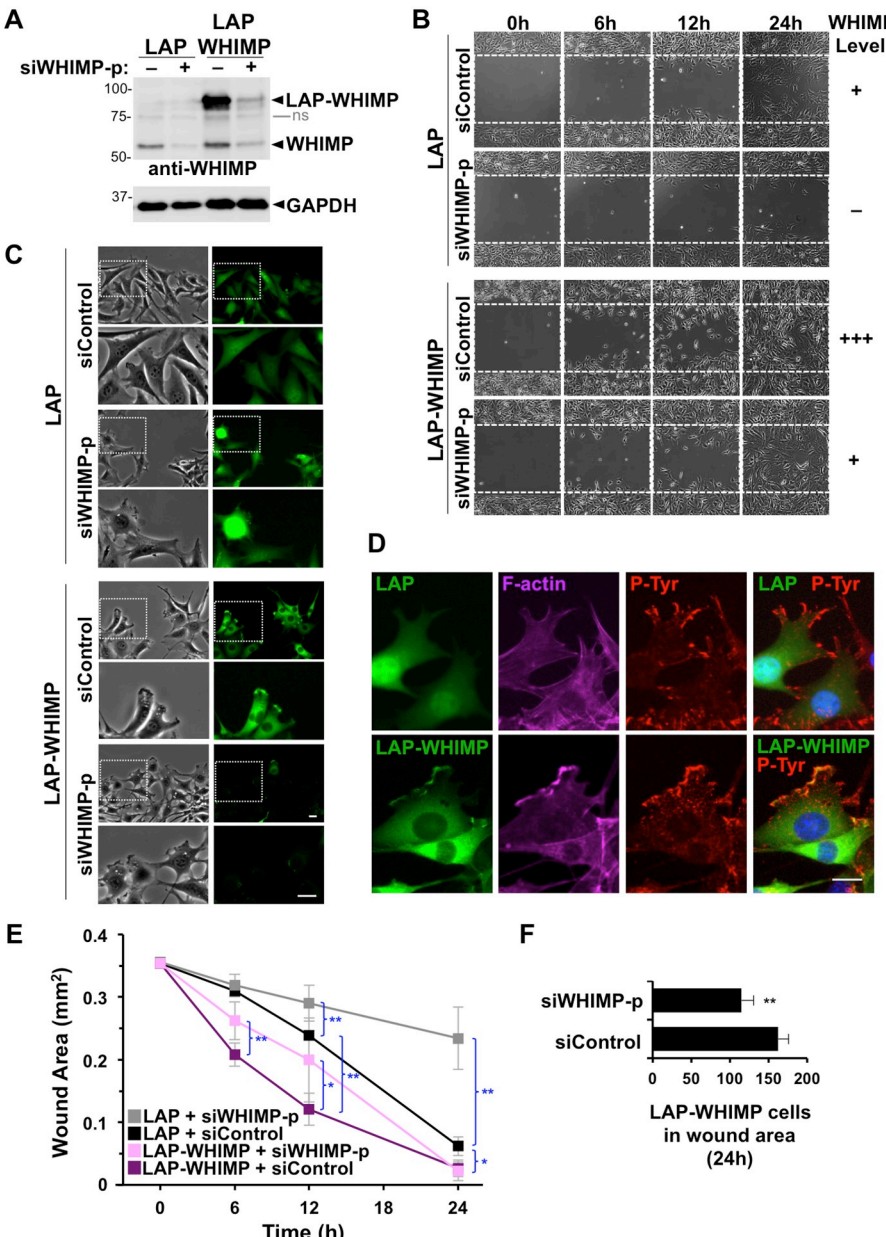

**Fig 7. WHIMP expression positively correlates with wound closure rates. (A)** NIH3T3 cell lines stably-encoding LAP or LAP-WHIMP were treated with a control siRNA (–) or a pool of siRNAs to WHIMP (+), induced to express the fusion proteins, subjected to SDS-PAGE, and immunoblotted for WHIMP and GAPDH. The tagged and endogenous versions of WHIMP are indicated. ns, non-specific. **(B)** Cells treated as in part A were grown in cell reservoirs separated by a 0.5mm barrier and analyzed by phase-contrast microscopy at 0, 6, 12, and 24 h after removing the barrier. **(C)** Cells treated as in part B were examined by fluorescence microscopy after removing the barrier. **(D)** Cells treated as in part B were fixed at the 6h timepoint, and stained with phalloidin (magenta), anti-phosphotyrosine antibodies (red), and DAPI (blue). Cells directed upwards toward the wound are shown. **(E)** The areas of barrier-created wounds that were generated as in part B were quantified. Each point represents the mean +/-SD from 3 experiments in which 2 wound areas were examined per condition. **(F)** Each bar represents the mean number of LAP-WHIMP cells present in the wound area at the 24h timepoint +/-SD from 3 experiments. $^{**}$ p<0.01, $^{*}$ p<0.05 (two-tailed t-tests). Scale bars, 20μm.

Similar to WHIMP depletions in NIH3T3 cells (Fig 7E), WHIMP depletions in B16-F1 cells resulted in less efficient filling of wound areas by 12h (Fig 8C).

In order to compare the wound healing defects of WHIMP knockdown cells with cells depleted of other motility factors, we used siRNAs against Brk1 (an essential component of the WAVE regulatory complex), N-WASP, or JMY. Immunoblotting revealed 70% depletions of WAVE1 and WAVE2 in samples receiving the Brk1 siRNA, and 70% or 50% depletions of N-WASP or JMY in samples receiving either of those siRNAs (Fig 8A). In competition-type assays, we compared the migration of control versus nucleation factor depleted cells into the same wound (Fig 8B). To characterize various aspects of movement, we captured images of wound areas at 5 minute intervals for 12h. In addition to measuring the remaining space unoccupied by cells (Fig 8D), we tracked individual cells in order to generate migration trajectories and quantify cell speeds, ruffling frequencies, protrusion rates, and protrusion widths (Fig 8E–8I). After 6h and 12h, siWHIMP-treated cells occupied 2-5-fold smaller wound areas than siControl-treated cells (Fig 8B and 8D). Slower wound healing was also observed in siBrk1-treated cells (Fig 8B and 8D), although by 12h these cells had filled more of the wound than the siWHIMP-treated samples (possibly due to indirect depletion of all 3 WAVE isoforms via Brk1 targeting being less efficient than direct WHIMP targeting). Cells treated with siRNAs to N-WASP or JMY had only minor effects on wound healing (Fig 8D, S6B Fig), consistent with previous observations suggesting that these factors contribute to movement in a limited number of cell types or environments [20, 79]. Thus, among the nucleation-promoting factors in B16-F1 cells with proposed functions at the leading edge, the WAVE subfamily members and WHIMP play the most important roles in 2-dimensional wound closure assays.

To gain deeper insights into why the WHIMP-depleted samples exhibited slower wound healing kinetics, we studied the motility properties of individual cells. In general, cells treated with WHIMP or Brk1 siRNAs showed more limited movements than control cells (Fig 8E). To explore whether this was due to a lack of directionality or slower cell motility, we measured directional persistence and speed. All cells exhibited a similar degree of linearity in their migration patterns (Fig 8E), although siWHIMP-a treated cells had slightly more meandering trajectories (S6C Fig). In contrast, more consistent differences in the speed of migration were uncovered when comparing WHIMP- or WAVE-depleted cells to control cells (Fig 8F). Mean motility rates for control cells were generally 12–16μm/h, whereas rates for siWHIMP-a, siWHIMP-b, and siBrk1-treated cells were 9μm/h, 4μm/h, and 7μm/h, respectively (Fig 8F). These results suggest that WHIMP depletion results in slower wound closure, at least in part because of a reduction in cell speed.

Since the WAVE proteins and WHIMP both promote motility and localize to membrane protrusions, we next compared the effects of their depletion on the protrusive capacities of B16-F1 cells. Targeting of either Brk1 or WHIMP lowered the frequency of cells with membrane ruffles from 90% to approximately 25–35% (Fig 8G). Moreover, the WAVE- or WHIMP-depleted cells exhibited more than 2-fold reductions in membrane protrusion speeds (Fig 8H) and possessed ruffle widths that were roughly 35% smaller than those in control cells (Fig 8I). Taken together, these data indicate that, like the WAVE proteins, WHIMP enables the formation of frequent, large, and fast-moving protrusions that promote motility.

Given that the WAVE proteins are established as key protrusion factors and that WHIMP expression can enhance membrane ruffling, we next tested whether the WAVEs contribute to the formation of WHIMP-induced ruffles by treating the LAP-WHIMP-overexpressing cells with siRNAs to Brk1 or to WHIMP as a control (S7A and S7B Fig). As expected, the majority of LAP-WHIMP-expressing cells had ruffles, while depletion of tagged and endogenous WHIMP abrogated this hyper-ruffling phenotype (S7C Fig). Indirect depletion of the WAVEs by targeting Brk1 also caused a significant decrease in the frequency of ruffling, although not

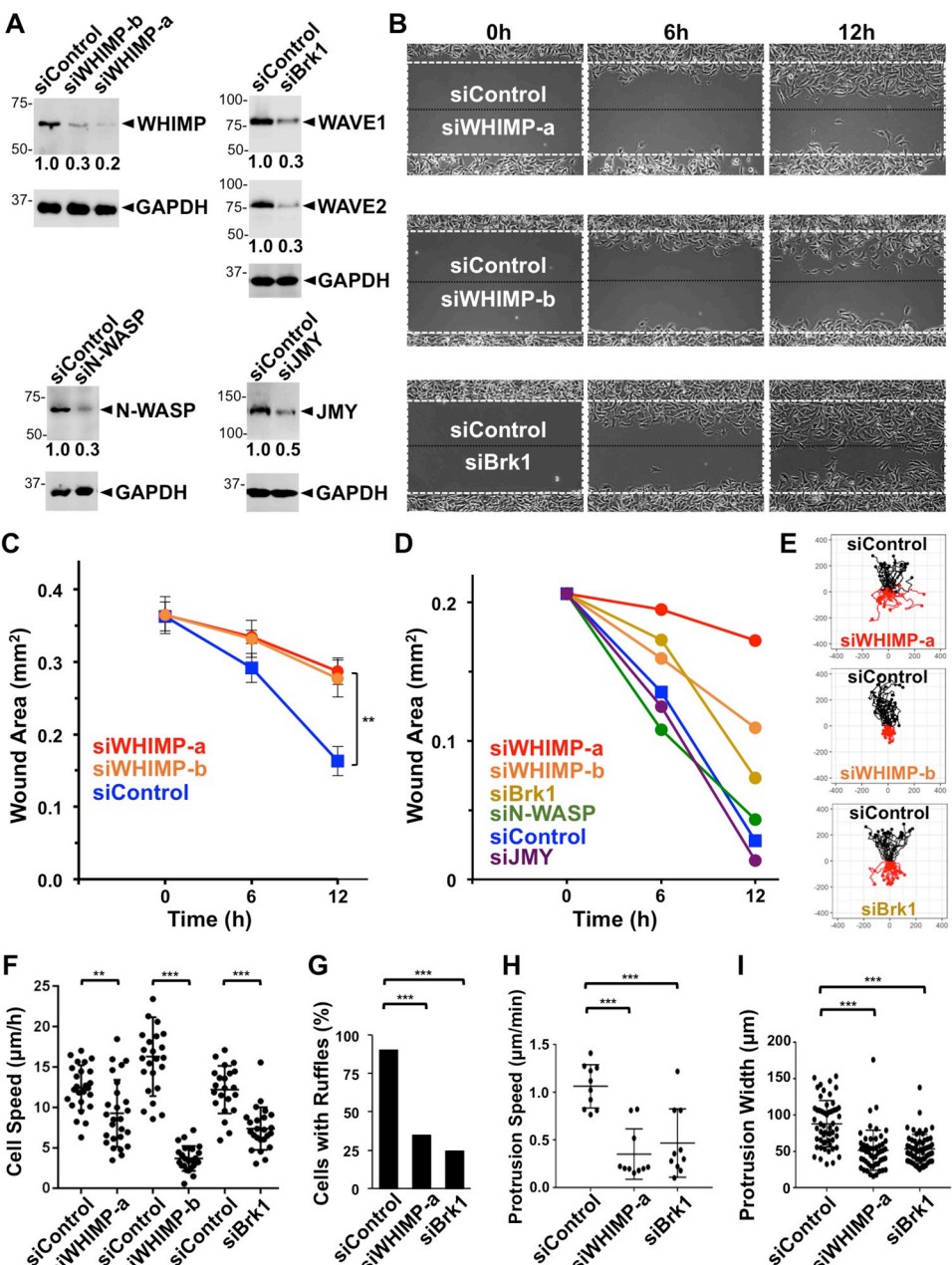

**Fig 8. WHIMP and the WAVE proteins both contribute to rapid cell motility and efficient protrusion. (A)** B16-F1 cells treated with either a control siRNA or siRNAs to WHIMP, Brk1, N-WASP, or JMY were lysed and subjected to SDS-PAGE and immunoblotting for WHIMP, WAVE1, WAVE2, N-WASP, JMY, or GAPDH. The average quantities of nucleation factor relative to GAPDH (shown beneath the nucleation factor panels) were determined by densitometry of 2–4 representative blots. **(B)** B16-F1 cells were treated with siRNAs as in part A, grown in cell reservoirs separated by a 0.5mm barrier, and analyzed by phase-contrast microscopy at 5min intervals for 12h after removing the barrier. Control cells are shown in the top chamber, and nucleation factor-depleted cells are shown in the bottom. The white dashed box encompasses the wound area at t = 0, and the black dashed line indicates the midpoint of the cell-free area. Wound healing movies appear in the Supporting Information section. **(C)** Cells grown in cell reservoirs separated by a 0.5mm barrier were analyzed by phase-contrast microscopy at 0, 6, and 12h after removing the barrier. Each point represents the mean +/-SD from 3 wounds as in Fig 7. **(D)** The areas of barrier-created wounds from assays described in part B were assessed at the indicated times. The siControl curve represents a typical sample from an siWHIMP competition experiment. **(E)** Individual cells within the wound areas from samples in B were tracked, and data from n≥20 randomly-chosen cells were plotted such that cell positions starting at t = 0 were centered at the origin. **(F)** Cells were tracked as in part E to determine individual motility rates. Each point represents the mean

speed of a randomly-chosen cell (n≥21). The horizontal line represents the overall mean +/-SD. **(G)** Cells treated with siRNAs as in part A were seeded onto glass-bottom wells and examined live by phase-contrast microscopy. The % of cells with ruffles was quantified. Each bar represents the % of cells (n≥85 per condition) with ruffles from a representative experiment. **(H)** Maximum protrusion speeds were calculated from samples in G based on the distance achieved by the largest continuous protrusion in the smallest amount of time within a 10min window of imaging as in Fig 5. Each point represents protrusion speed from one cell (n = 10), and horizontal lines are their mean +/-SD. **(I)** Cells treated with siRNAs as in part A and seeded as in part G were fixed and examined by phase-contrast microscopy. Each point represents the maximum protrusion width from one cell (n≥50), and the horizontal line represents their mean +/-SD. *** p<0.001, ** p<0.01 (ANOVA with Tukey post-test or Fisher's exact test in panel G).

nearly to the same extent as blocking WHIMP expression (S7C Fig). Since WAVE function classically requires G-protein-mediated signaling from Rac1, we tested if suppressing Rac activity could also reduce WHIMP-mediated ruffling. Indeed, treatment of cells with the Rac inhibitor Ehop-016 diminished WAVE1 localization to the plasma membrane and decreased the fraction of WHIMP-overexpressing cells with ruffles (S7D and S7E Fig). Thus, the potent WHIMP-mediated induction in ruffling can be partially suppressed by disrupting Rac/WAVE signaling.

## WHIMP enhances Src-family tyrosine kinase signaling to promote membrane ruffling

Our observations that WHIMP WCA-mediated Arp2/3 activation stimulates membrane protrusion, that multiple nucleation factors localize to WHIMP-induced ruffles, and that Rac/WAVE contribute to these protrusions together imply that WHIMP's functions in movement can be driven directly by its nucleation-promoting activity, but also amplified indirectly by a distinct signaling activity. Because tyrosine kinases provide important signals for protrusion and adhesion [43–45, 47], and phosphotyrosine staining was broadly and intensely localized to the front of migrating LAP-WHIMP cells (Fig 7D), we next characterized the tyrosine phosphorylation patterns in LAP-, LAP-WHIMP-, and LAP-ΔWCA-expressing cells. In control LAP cells, phosphotyrosine staining was apparent in expected punctate or wedge-like structures resembling focal adhesions in the cell periphery (Figs 7D and 9A). LAP-WHIMP cells also had phosphotyrosine-positive focal contacts, but more noticeably, these cells possessed a strong accumulation of phosphotyrosine staining in their plasma membrane ruffles (Figs 7D and 9A). LAP-WHIMP showed co-localization with phosphotyrosines in the ruffles (Fig 9A), and line-scans of multiple cells expressing LAP versus LAP-WHIMP indicated both an increase in the phosphotyrosine intensity in WHIMP-induced protrusions, and high levels of F-actin in such structures (Fig 9B; S8A and S8B Fig). Similar results were obtained when cells were stained with antibodies to the focal adhesion protein vinculin (S8A and S8B Fig). Confocal Z-stacks confirmed the phosphotyrosine staining pattern, showing a strong overlapping peak of localization among WHIMP, phosphotyrosine, and F-actin (Fig 9B). In contrast, cells expressing the LAP-WHIMP(ΔWCA) mutant, which form less prominent ruffles, had correspondingly narrower phosphotyrosine staining patterns (Fig 9A). Downstream of tyrosine kinase activation, Rac1 can induce membrane ruffling via WAVE recruitment, but protrusions triggered by direct expression of constitutively active Rac1(Q61L) did not contain the broad phosphotyrosine staining pattern that was characteristic of WHIMP-expressing cells (Fig 9A). These results suggest that abundant tyrosine phosphorylation is not a prevalent feature within all ruffles, but may be specifically elicited by WHIMP.

To better define how tyrosine phosphorylation is altered upon WHIMP overexpression, we examined the levels and profiles of phosphoproteins by immunoblotting extracts isolated from cells expressing LAP, LAP-WHIMP, and LAP-ΔWCA. First, compared to LAP control cells,

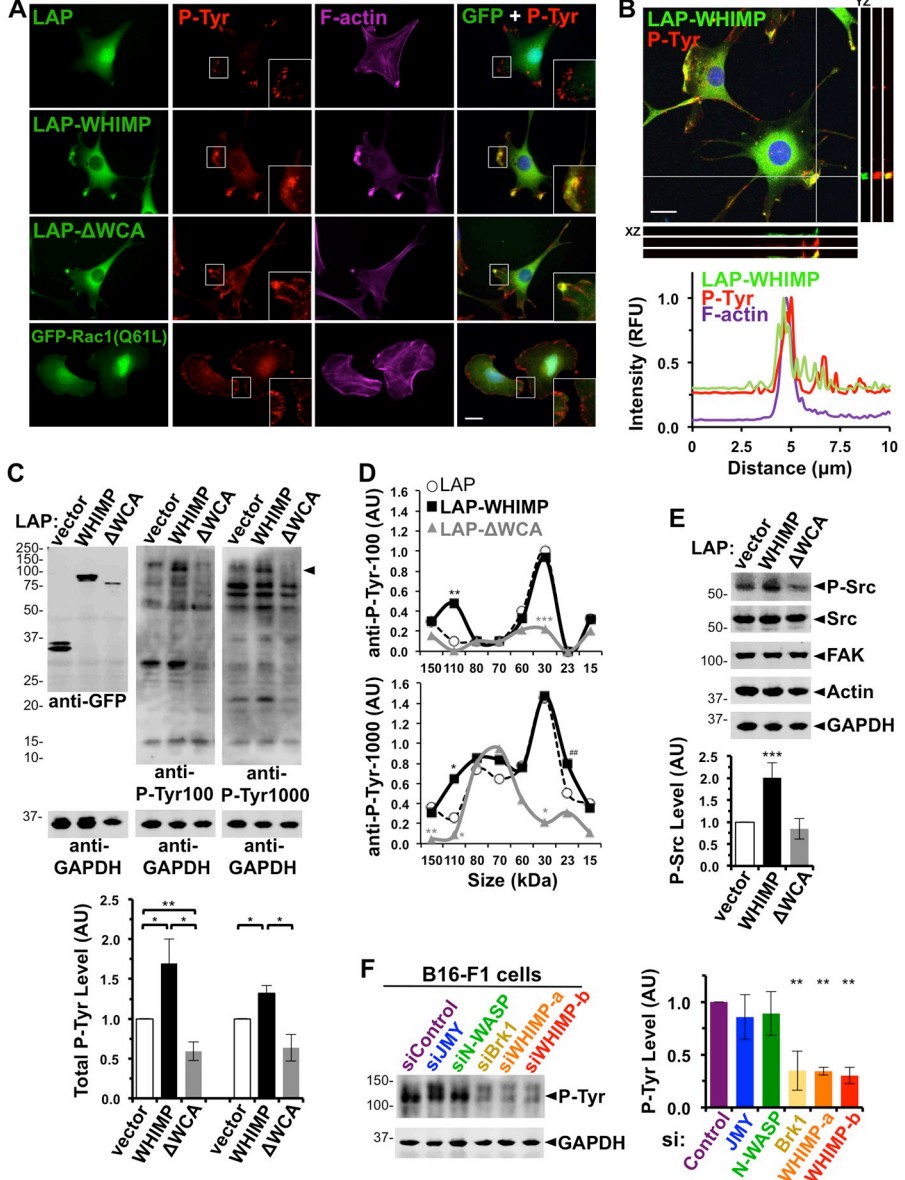

**Fig 9. WHIMP increases tyrosine phosphorylation and Src-family kinase activation. (A)** NIH3T3 cell lines expressing LAP, LAP-WHIMP, or LAP-WHIMP(ΔWCA), or transiently transfected with a plasmid encoding GFP-Rac1(Q61L) were fixed and stained with antibodies to phosphotyrosine (P-Tyr) (red), phalloidin to visualize F-actin (magenta), and DAPI to label DNA (blue). Insets highlight protrusions, including a WHIMP-associated ruffle with a broad enrichment of phosphotyrosine staining compared to wedge-like phosphotyrosine staining in other cells. **(B)** LAP-WHIMP-expressing cells stained as in A were subjected to confocal microscopy. Maximum intensity projections are shown in the large panel, and adjacent orthogonal views of YZ and XZ planes are indicated by the two gray lines. The plot profiles depict pixel intensities of LAP-WHIMP, phosphotyrosine, and F-actin along a 10μm line drawn near the intersection of the two gray lines. **(C)** Cell lines expressing LAP, LAP-WHIMP, or LAP-WHIMP (ΔWCA) were subjected to SDS-PAGE and immunoblotted with antibodies to GFP and GAPDH, and two different antibodies (P-Tyr100, P-Tyr1000) to phosphotyrosine. The arrowhead indicates the position of a ~110kDa phosphoprotein that is enriched in LAP-WHIMP-expressing cells. Total phosphotyrosine levels per lane were normalized to GAPDH and actin, and were quantified in the lower graph. Each bar represents the mean +/-SD from 5 experiments. **(D)** Phosphotyrosine bands at the specific sizes indicated on the X-axis were quantified as in panel C. Each point represents the mean intensity from 4–6 experiments. Color-coded asterisks indicate significant differences from LAP control samples. *** p<0.001, ** p<0.01, * p<0.05 (two-tailed t-tests). ##, p<0.01 between LAP-WHIMP, LAP-ΔWCA. **(E)** Cells processed as in C were immunoblotted for active Src P-Tyr416 (P-Src), total Src, FAK, actin, and GAPDH. The samples in E are the same as those used in the middle panel of C. Active P-Src levels were normalized to total Src and quantified. Each bar represents the mean +/-SD from 5 experiments. **(F)** B16-F1 cells

treated with either a control siRNA or siRNAs to JMY, N-WASP, Brk1, or WHIMP were lysed and subjected to SDS-PAGE and immunoblotting for phosphotyrosine (P-Tyr100) and GAPDH. Phosphotyrosine levels in the 100-150kDa range were normalized to GAPDH and actin, and were quantified in the adjacent graph. Each bar represents the mean +/-SD from 3 experiments. *** p<0.001, ** p<0.01, * p<0.05 (ANOVA with Tukey post-test). Scale bars, 20μm.

the overall phosphotyrosine content in LAP-WHIMP-expressing cells was 35–70% higher, whereas in LAP-ΔWCA cells it was 35–40% lower (Fig 9C). Second, compared to LAP cells, LAP-WHIMP cells displayed a significantly different phosphoprotein profile when probed with anti-phosphotyrosine antibodies (Fig 9C and 9D). Most notable was the appearance of a prominent ~110kDa phosphoprotein in the WHIMP-overexpressing samples (Fig 9C), which was confirmed to be present at 3-5-fold higher levels than in LAP control cells by densitometric analysis (Fig 9D). LAP-ΔWCA cells lacked this phosphoprotein (Fig 9C and 9D), and the other most distinguishing feature of these mutant-expressing cells was the loss of a ~30kDa phosphoprotein (Fig 9C and 9D).

Although the specific identities of these differentially-phosphorylated proteins remain unknown, we next sought to determine which kinases might be activated in the LAP-WHIMP-expressing cells to give rise to the overall increase in tyrosine phosphorylation. Src-family kinases (SFKs) regulate cell-matrix adhesion, promote cell migration, and can increase actin assembly by binding or phosphorylating N-WASP or the WAVE proteins [53, 55, 56, 65]. To evaluate SFK activity in our cell lines, we immunoblotted protein extracts with antibodies for Src and for an active form of Src which is phosphorylated on Tyr416. While the total levels of Src and one of its binding-partner kinases, FAK, were equivalent in all 3 NIH3T3 cell derivatives, active Src P-Tyr416 levels were doubled in LAP-WHIMP-expressing cells compared to LAP or LAP-ΔWCA cells (Fig 9E). Src P-Tyr416 levels appeared normal in GFP-N-WASP- or GFP-Cortactin-expressing cells (S6D Fig). We next examined the localization of Src P-Tyr416 in LAP-WHIMP cells. Src P-Tyr416 was abundant throughout WHIMP-induced F-actin-rich ruffles but exhibited a fairly normal adhesion-type localization in LAP-ΔWCA cells (S8C Fig). These data suggest that overexpression of full-length WHIMP is sufficient to increase the activity of Src and alter its localization without affecting its overall abundance.

To test if endogenous WHIMP is necessary for SFK activation, we immunoblotted protein extracts from B16-F1 cells treated with siRNAs to different nucleation factors. While targeting of N-WASP or Jmy did not substantially impact phosphotyrosine levels, treatment with independent WHIMP siRNAs resulted in less phosphotyrosine staining (Fig 9F). Targeting of Brk1 to promote WAVE-depletion also yielded cell extracts with less phosphoproteins (Fig 9F). Thus, both WHIMP and the WAVEs are necessary for maintaining normal levels of tyrosine phosphorylation in cells.

Lastly, to evaluate the functional significance of SFK activation in WHIMP-associated membrane ruffling, we treated LAP-WHIMP-expressing cells with 2 distinct SFK inhibitors, PP2 or Saracatinib [83, 84]. Compared to DMSO-treated control cell populations, which had numerous large ruffles, cells treated with PP2 for 2 hours displayed fewer ruffles, and showed reductions in both phosphotyrosine and F-actin staining at the cell periphery (Fig 10A and 10B; S8D Fig). As predicted, these PP2-dependent phenotypes were accompanied by a diffusion of Src P-Tyr416 and FAK immunofluorescence at the cell periphery (Fig 10B; S8D Fig). Quantification of ruffling demonstrated that LAP-WHIMP cells treated with PP2 had half as many ruffles as cells treated with DMSO (Fig 10C), and the LAP-WHIMP-associated ruffles that remained were reduced to the same size as those observed in LAP control cells (Fig 10D). Like treatment with PP2, incubation with Saracatinib caused fewer LAP-WHIMP cells to have

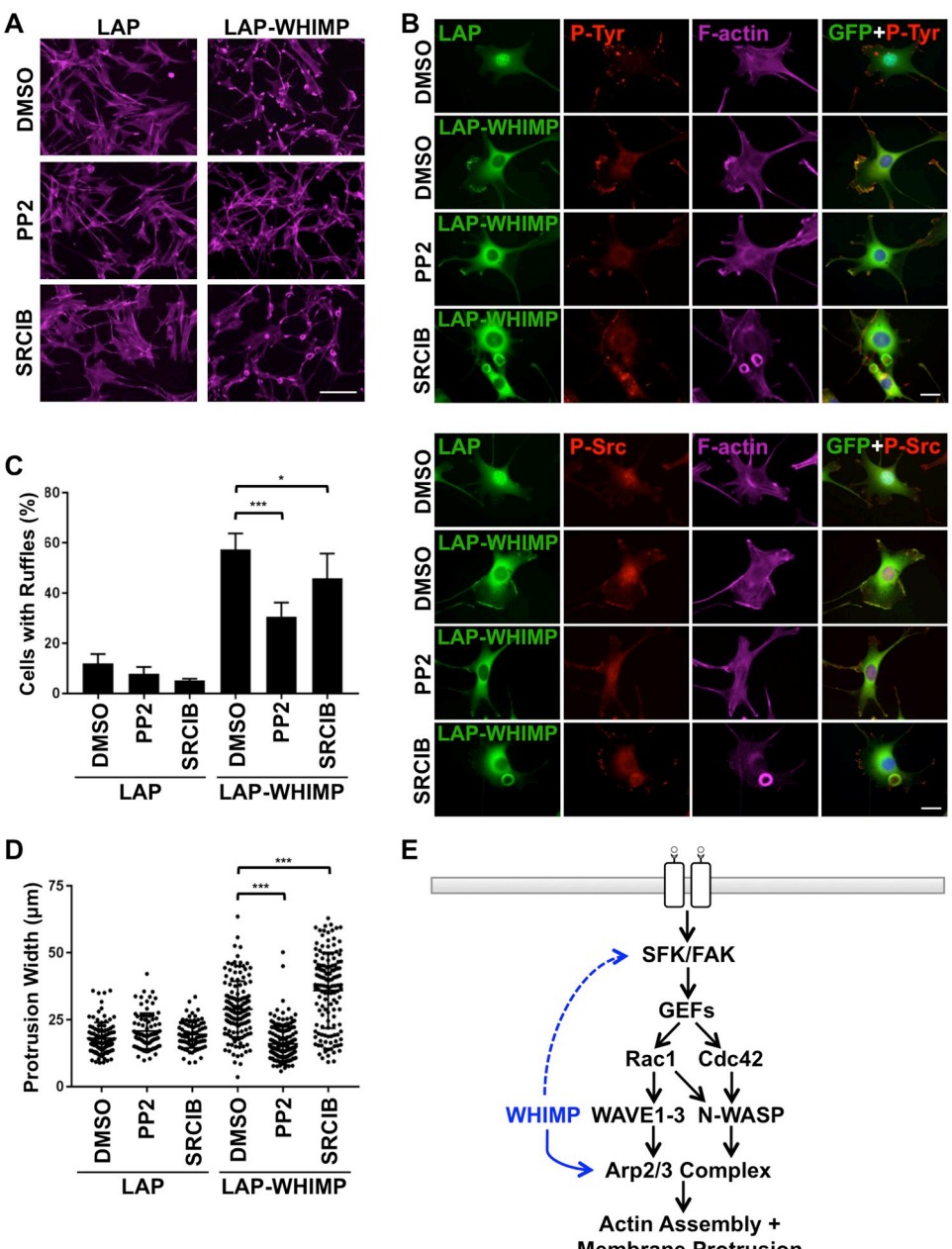

**Fig 10. WHIMP-induced membrane ruffling relies on Src-family kinase activity. (A)** NIH3T3 cell lines expressing LAP or LAP-WHIMP were treated with DMSO or with 20μM PP2 or Saracatinib (SRCIB) for 2h, fixed, and stained with phalloidin to visualize F-actin (magenta). Scale bar, 100μm. **(B)** Cells treated as in A were stained with antibodies to phosphotyrosine or active Src P-Tyr416 (P-Src; red), phalloidin to visualize F-actin (magenta), and DAPI to label DNA (blue). Scale bars, 20μm. **(C)** The % of cells with ruffles was quantified from experiments performed in A-B. Each bar represents the mean % of cells (n = 200–350 per condition) with ruffles from n = 4–6 coverslips +/- SD pooled from 3 experiments. **(D)** Membrane ruffle widths were measured in LAP- or LAP-WHIMP-expressing cells (n = 76–175 per condition pooled from 3 experiments). Each point represents a single cell, and the horizontal line represents their mean +/-SD. *** p<0.001, * p<0.05 (ANOVA with Tukey post-test). **(E)** A model for newly-described WHIMP functions (blue) in direct Arp2/3 activation (solid line) and in promoting SFK activity (dashed line) during actin assembly, membrane protrusion, and cell motility is shown.

ruffles (Fig 10A). However, this reduction was fairly modest (Fig 10C). Unexpectedly, Saracatinib had striking effects on the pattern of membrane ruffling, as the LAP-WHIMP-expressing population became dominated by cells with circular ruffles containing intense F-actin staining (Fig 10A and 10B; S8D Fig). These structures were not found in the LAP-expressing samples (Fig 10A–10C). Moreover, despite the presence of LAP-WHIMP and F-actin in the Saracatinib-induced circles, these structures were not enriched for phosphotyrosine, Src P-Tyr416, or FAK staining (Fig 10A and 10B; S8D Fig). Thus, while PP2 inhibits WHIMP-mediated ruffling generally, Saracatinib causes a remarkable shift from peripheral to dorsal ruffling. While the molecular basis of such differences (e.g., target specificity, intracellular drug distribution) are unclear, the results from these pharmacological treatments are nevertheless consistent with Src-family tyrosine kinases acting as key modulators of WHIMP-induced membrane protrusion.

## Discussion

Nucleation-promoting factors from the WASP family assimilate upstream signals and activate the Arp2/3 complex to drive actin assembly. The physiological importance of these factors is underscored by the observations that murine N-WASP, WAVE1, WAVE2, and WASH gene knockouts result in lethality [19, 20, 22, 29], that mutations in human *WAS* give rise to immunodeficiencies [18, 85], and that a mutation in *WHAMM* is found in patients with a neurodevelopmental/renal disorder [38, 86]. All family members have been shown to influence cell motility, with at least 6 playing roles in membrane protrusion. Given their significance in cytoskeletal functions, development, and disease, we investigated whether additional proteins within the WASP family exist. Here we characterized WHIMP as a new actin assembly factor that promotes activation of the Arp2/3 complex as well as Src-family kinases to induce protrusion and enhance cell motility.

Many of the WASP-family proteins are found universally across mammalian species, but current bioinformatic analyses suggest that WHIMP is present in rodents plus a subset of other mammals. Sequence comparisons indicate that WHIMP most closely resembles WAVE1, and since WHIMP possesses a WCA domain, we investigated its role in actin assembly *in vitro* and in cells. We found that WHIMP is an Arp2/3-dependent actin nucleation-promoting factor, but has much weaker activity than other family members. Our pyrene-actin assembly assays also provide one of the broadest comparisons of WCA domain activities. Based on these results, we can place them into a potency hierarchy where WASP, WAVE2 > N-WASP, JMY > WHAMM > WASH > WHIMP. In cells, WHIMP and its WCA domain were also able to cause actin assembly, but again with a relatively low potency.

Similar to other nucleation factors that can be recruited to the plasma membrane, tagged WHIMP co-localized in membrane protrusions with the Arp2/3 complex. In fact, WHIMP localization overlapped significantly with WAVE1, WAVE2, and N-WASP in ruffles. WHIMP was not only found within these structures, but increasing WHIMP expression dramatically enhanced the frequency and size of ruffles. In contrast, expression of a WHIMP mutant lacking its WCA domain failed to increase actin assembly, ruffle size, or protrusion speed. Consistent with these results, treatment of LAP-WHIMP-expressing cells with an Arp2/3 complex inhibitor diminished ruffling. Together, these data provide strong evidence that the WHIMP WCA domain promotes Arp2/3-dependent actin assembly to drive the formation and movement of membrane ruffles.

Because membrane protrusion is a key step in cell motility, and WHIMP is present in ruffles like other WASP-family proteins, we examined its function in wound healing cell migration assays. NIH3T3 cells overexpressing LAP-WHIMP showed WHIMP localization at the

leading edge and faster wound closure. In contrast, WHIMP depletion in both NIH3T3 and B16-F1 cells led to slower rates of closure. These results add WHIMP to the list of WASP-family members that can regulate actin dynamics at the front of cells to enable cell motility.

A large body of evidence previously indicated that WASP, N-WASP, WAVE1-3, and JMY can all promote actin polymerization, protrusion, and motility [4, 5, 27]. However, deciphering the precise roles of these 6 factors has been challenging and is accompanied by some debate. Depending on the cell type and environmental context, motility may involve the activities of different combinations of these proteins at the leading edge. Experiments from long ago [25] through current times [87] have established that the WAVEs play major roles in actin-driven protrusion at the front of the cell, although the 3 isoforms may engage in non-redundant functions. For instance, studies in mouse embryonic fibroblasts (MEFs) showed that WAVE2 deficiency impairs the formation of peripheral membrane ruffles, while WAVE1 deficiency inhibits the formation of dorsal ruffles [80, 81]. Moreover, WAVE2 knockout MEFs lack lamellipodia, whereas WAVE1 knockouts have rapidly protruding but short-lived lamellipodia [22, 88]. In addition, WASP/N-WASP can be found at the front of different types of motile cells [89–91], but their functions in protrusion and motility are not as well accepted as those of the WAVEs. In multiple blood cell lineages, WASP was shown to be important for protrusion, motility, and/or chemotaxis (summarized in [27]), although some of these defects might be an indirect result of developmental deficiencies or transcriptional alterations [92, 93]. The significance of the ubiquitously-expressed N-WASP in protrusion is less clear. N-WASP appears to be critical for 3D invasion [94–96], and N-WASP may contribute to 2D motility by some cells [89], but knockouts in mouse fibroblasts and knockdowns in several cancer cell lines suggest that N-WASP is dispensable for lamellipodial assembly [20, 97, 98]. On the other hand, phylogenetic and functional analyses indicate that the presence of both the WAVE and WASP subfamilies is necessary for fast pseudopod-based amoeboid-like motility [27]. Finally, initial studies examining JMY function showed a positive relationship between JMY expression levels and wound healing rates in human U2OS [42] and MCF-7 cells [99]. However, similar effects were not observed upon JMY depletion in MEFs [79], and the significance of JMY in protrusion is unsettled. Overall, these observations demonstrate that plasma membrane protrusion and cell motility involves complex sets of factors and regulatory mechanisms.

Our study sheds further light on such processes by testing roles for each of these WASP-family members in motility using RNAi-mediated transient knockdowns. In B16-F1 cells, as expected, depletion of the WAVEs by targeting Brk1 slowed wound healing. In contrast, knockdown of N-WASP or JMY did not have a substantial effect on wound closure. The defect in wound filling by both the WAVE- and the WHIMP-depleted cells appeared to be a consequence of decreased cell speed, a phenotype accompanied by smaller and slower moving membrane protrusions. Thus, at least in this melanoma cell line, 2D migration relies primarily on the WAVEs and WHIMP without major contributions from N-WASP or JMY. Interestingly, WHIMP-induced ruffling in NIH3T3 cells could be diminished by targeting the WAVE complex or its upstream activators, the Rac G-proteins. These results suggest that some degree of functional cooperation exists between WHIMP and the WAVEs during protrusion. The extent to which these different nucleation-promoting factors work sequentially, in parallel, or otherwise remains to be determined.

One clue as to how WHIMP might collaborate with other nucleation factors in protrusion and motility came from staining cells for phosphotyrosines. These studies revealed that WHIMP co-localized with phosphotyrosine staining in ruffles and that WHIMP overexpression resulted in a broader distribution of phosphotyrosines at the leading edge. Upon further investigation, compared to cells expressing endogenous levels of WHIMP, the overall abundance of tyrosine phosphorylated proteins was higher in WHIMP-overexpressing cells and

lower in WHIMP-depleted cells. The increase in phosphoprotein levels was coupled with greater amounts of active Src-family kinases, and inhibition of these kinases perturbed ruffling, indicating that continuous SFK activation is necessary to maintain WHIMP-mediated protrusions.

The positive association between WHIMP expression and Src activation could explain the localization of WAVE1, WAVE2, and N-WASP in WHIMP-induced ruffles, as SFK signaling can occur upstream of the G-proteins that recruit the WAVEs and WASPs to the plasma membrane [43–45]. In addition, such kinases could directly enhance actin assembly in protrusions, because binding or phosphorylation by tyrosine kinases results in activation of the WAVEs and WASPs [52–56, 65–68]. Our results therefore suggest that WHIMP enhances membrane protrusion via two mechanisms: (i) by directly stimulating the Arp2/3 complex to drive actin assembly and (ii) by promoting the activation of upstream kinases that sustain the signals for stimulating multiple nucleation factors (Fig 10E). Interestingly, previous examinations of the atypical Arp2/3 activators Cortactin and WISH/DIP1/SPIN90 have shown that they both not only activate Arp2/3, but can also influence upstream signaling from G-proteins [100–102]. Thus, positive or negative feedback pathways may be a commonly-held feature of nucleation factors.

In the future, it will be important to gain a better understanding of the mechanisms governing WHIMP's activation of SFKs and identify the phosphorylated targets that mediate the enhancement of ruffling and motility. The characterization of WHIMP's potential binding-partners, including SFKs, other tyrosine kinases, or even phosphatases should begin to clarify such mechanisms, while phosphoproteomic analyses of cells expressing different levels of WHIMP will help reveal the modified targets. Another key area to address is how the different WASP-family members are coordinated, and determine whether the spatial organization of different nucleation factors in lamellipodia has functional consequences. For example, it is tempting to speculate that the more potent Arp2/3 activators like WAVE2 drive particularly explosive foci of protrusion, while weaker factors like WHIMP result in comparably slower but more persistent extensions. It will also be crucial to establish how cell adhesion behaviors occur relative to protrusion. Finally, the reason why only a subset of mammals appears to encode WHIMP is a lingering question. Perhaps other proteins can substitute for WHIMP's actin assembly and kinase stimulating effects in some organisms or cell types. Two candidates that could fulfill these roles are WISH/DIP1/SPIN90, which can affect Src-dependent feedback during growth factor signaling [102], and the Formin-family nucleator mDia1, which promotes SFK activation in pathogen-induced membrane protrusions [103]. Our current findings lay the foundation for answering these questions about the functions of multiple actin assembly proteins and for gaining a better understanding of WHIMP as a new WASP-family member.

## Materials and methods

### Ethics statement

Research with biological materials in the Campellone Lab was approved by the UConn Institutional Biosafety Committee (#759C). This study did not include human subjects research or research with live animals. Mouse cell lines were acquired from the American Type Culture Collection (ATCC) as described below. Mouse tissues were a gift from Michael O'Neill's lab and were collected with approval from the UConn IACUC (#A19-047).

### Plasmids, bacteria, viruses, and cells

Plasmids are listed in Supplemental S1 Table. For cloning, DNA fragments were amplified from cDNA templates by PCR, digested, and ligated into the appropriate restriction enzyme

sites. Plasmids were maintained in *E.coli* XL-1 Blue (Stratagene). Bacmids were maintained in DH10Bac (Invitrogen). All bacteria were grown at 37˚C. Baculoviruses were generated using the Bac-to-Bac system (Invitrogen). Sf9 insect cells were grown in ESF921 media (Expression Systems) plus antibiotic-antimycotic at 28˚C. NIH3T3, Neuro2a, HT22, and B16-F1 cells (ATCC) were grown in Dulbecco's Modified Eagle's Medium (DMEM), plus 10% fetal bovine serum (FBS) and antibiotic-antimycotic at 37˚C in 5% $CO_2$.

## Tissue preparation

Brain, heart, kidney, liver, and testis from two 5-month-old male C57/B6 mice were washed with phosphate-buffered saline (PBS) and frozen in liquid nitrogen. Tissue extracts were prepared by resuspending organs in tissue lysis buffer (10mM Tris pH7.4, 150mM NaCl, 1mM EDTA, 1% Triton X-100, and a protease inhibitor cocktail consisting of 1mM PMSF and 10μg/ml each of aprotinin, leupeptin, pepstatin, and chymostatin) and performing three 30s pulses with a Polytron PT-1200 tissue homogenizer. The lysates were then clarified by centrifugation at 10,000x*g* for 30min at 4˚C. Extract concentrations were measured using Bradford assays (Bio-Rad).

## RT-PCR

Total RNA from NIH3T3, HT22, and Neuro2a cells was extracted using the Purelink minikit (Ambion). 2μg of RNA was used as substrate for first strand cDNA synthesis with Superscript III RT (Invitrogen). For PCR, 4μl of synthesized cDNA mixtures were used in 25μl reactions with Taq polymerase (New England Biolabs). For normalization, β-actin was used as a control. Sequences of PCR primers were ATGGAGAATGATAAGAGTGAAGAACAAAGG & GGA-TATGTTCTTTCAAGAGTTGACACTGTA for WHIMP, and GCTCGTCGTCGACAAC GGCT & GGTCATCTTCTCGCGGTTGG for β-actin.

## Transfection

Prior to DNA and RNA transfections, NIH3T3 or B16-F1 cells were grown in 24- or 6-well plates for 24h. For expression of fluorescently-tagged proteins, cells were transfected with 250-1500ng of DNA using LipofectamineLTX with Plus reagent (Invitrogen). For RNA interference experiments, cells were transfected with 50nM siControl RNAs (Ambion), siWHIMP Stealth siRNAs (MSS210515, MSS210516, MSS277782), siControl1 RNAs (Sigma), siBrk1 (00028245), siN-WASP (00117645), or siJMY (00043711) using RNAiMAX (Invitrogen). Cells were incubated in growth media for 24h after DNA transfections and 48h after siRNA transfections. Cells grown in 6-well plates were collected and processed for immunoblotting, and cells grown on 12mm glass coverslips in 24-well plates were fixed using paraformaldehyde.

## Stably-transfected cell lines

NIH3T3 cells cultured in 6-well plates were transfected with 3–4μg of linearized plasmid encoding LAP- or mCherry-WHIMP derivatives. 24h later, the cells were transferred to growth media containing 800μg/mL G418 for 48h. Cells were then reseeded into T-75 flasks, allowed to reach ~80% confluency, induced in media containing 10mM sodium butyrate for 16-18h, and subjected to fluorescence activated cell sorting (FACS). GFP- or mCherry-positive cells were grown in G418-containing media for at least one week before being returned to normal growth media and subjected to experimental manipulations. Cells transfected with LAP-WHIMP(ΔWCA) proliferated more slowly than other lines. Upon subsequent inductions, the penetrance of ΔWCA expression was also lower than other constructs, and its level of fluorescence declined over time, suggesting that this fusion protein is not well tolerated by cells.

Transiently expressed GFP-WAVE1 and GFP-WAVE2 induce cytoplasmic actin assembly and sequestration of the Arp2/3 complex [10, 33] presumably because the overexpressed fusion proteins are present at levels too high to be incorporated into the WAVE regulatory complex. Ectopic actin assembly and Arp2/3 sequestration was not apparent upon expression of GFP-N-WASP or GFP-Cortactin, so cell lines stably encoding those two constructs were used as controls for ruffling and phosphotyrosine-related assays.

## Immunoblotting

To prepare whole cell extracts, cells were washed with PBS, collected, pelleted via centrifugation, and lysed in 50mM Tris-HCl (pH 7.6), 50mM NaCl, 1% Triton X-100, plus PMSF, aprotinin, leupeptin, pepstatin, and chymostatin. $Na_3VO_4$ and NaF were included at 1mM in experiments involving phosphotyrosines. Protein samples were mixed with SDS-PAGE sample buffer, boiled, and subjected to SDS-PAGE before transfer to nitrocellulose membranes (GE Healthcare). Membranes were blocked in PBS with 5% milk before being probed with antibodies. Affinity-purified chicken anti-WHIMP antibodies were generated against mouse WHIMP amino acids 9–21 (Aves Labs). Other antibodies were rabbit anti-MBP [70], mouse anti-GFP (Santa Cruz, sc9996), rabbit anti-mCherry (Abcam, ab167453), mouse anti-tubulin (Developmental Studies Hybridoma Bank, E7), mouse anti-GAPDH (Invitrogen, AM4300), mouse anti-β-actin (Proteintech, 66009-1-Ig), rabbit anti-WAVE1 (Abcam, ab50356), rabbit anti-WAVE2 (Cell Signaling, 3659), guinea pig anti-N-WASP [32], rabbit anti-JMY (Proteintech, 25098-1-AP), mouse/rabbit anti-phosphotyrosine P-Tyr100/P-Tyr1000 (Cell Signaling, 9411/8954), rabbit anti-Src (Cell Signaling, 2109), rabbit anti-Src-P-Tyr416 (Cell Signaling, 6943), and rabbit anti-FAK (Cell Signaling, 13009). Secondary antibodies were conjugated to IRDye 800CW or IRDye680LT (LI-COR), or horseradish peroxidase (GE Healthcare). Images were captured using a LI-COR Odyssey Fc Imaging System. Band intensities were determined using LI-COR Image Studio software. Quantities of proteins-of-interest (e.g., siRNA targets, phosphoproteins, etc.) were normalized to GAPDH, tubulin, and/or actin loading controls.

## Protein expression and purification

Sf9 cells were infected with baculoviruses as described previously [70]. Cells were collected 72h after infection and centrifuged at 3,500rpm for 12min at 4˚C in a Fiberlite F12-6x500 fixed-angle rotor. Cell pellets were resuspended in MBP buffer (20mM Tris pH7.6, 250mM NaCl, 100mM KCl, 5% glycerol, 1mM EDTA, 1mM DTT, plus PMSF, aprotinin, leupeptin, pepstatin, and chymostatin), and frozen using liquid nitrogen. For lysis, infected cell suspensions were freeze-thawed in the presence of 0.1% Triton X-100, sonicated at 60% power for 35s three times using a Fisher dismembranator, and centrifuged at 35,000rpm for 20min at 4˚C in a Beckman Coulter 70.1Ti rotor. Supernatants were passed through 0.45μm filters and mixed with amylose resin (New England Biolabs) to isolate the MBP-tagged proteins. Bound proteins were eluted in MBP buffer supplemented with 10mM maltose. Purified proteins were then dialyzed into pyrene-actin assay control buffer (20mM MOPS pH7.0, 100mM KCl, 2mM $MgCl_2$, 5mM EGTA, 1mM EDTA, 10% glycerol) using Slide-A-Lyzer Mini Dialysis Devices (Thermo Scientific) and concentrated using 30kDa MWCO Amicon Ultra-4 centrifugal filter units (Millipore). Protein concentrations were measured using Bradford assays.

## Pyrene-actin assembly assays

Pyrene-actin assembly assays were performed similar to experiments described previously [34, 38]. Actin (Cytoskeleton Inc.) was resuspended in G-buffer (5mM Tris pH8.0, 0.2mM $CaCl_2$, 0.2mM DTT, 0.2mM ATP) and subjected to gel filtration chromatography using a

Superdex200 column (GE Healthcare) to remove small filaments. Actin (2μM; 7% pyrene-labeled) was then polymerized in the presence of 20nM bovine Arp2/3 complex (Cytoskeleton Inc.) plus MBP-tagged fusion proteins in control buffer supplemented with 0.2mM ATP and 0.5mM DTT. Pyrene fluorescence was measured using a Horiba Jobin Yvon Fluorolog-3 spectrofluorimeter capable of multi-wavelength excitation/detection and equipped with a four-position sample changer. Weak nucleation-promoting activity for full-length WHIMP and WHIMP(WCA) was observed in at least 4 experiments with 2 separate protein preps. Polymerization parameters were calculated similarly to previous studies [13, 104] by measuring the slopes of pyrene curves at half of the maximal F-actin concentration using Microsoft Excel. Graphs for maximum actin polymerization rates and times to half maximal polymer were generated using GraphPad Prism.

## Fluorescence microscopy

NIH3T3 cells cultured on glass coverslips in 24-well plates were fixed using 2.5% or 3.7% para-formaldehyde, and permeabilized with 0.1% Triton X-100 diluted in PBS. Antibody staining was performed using mouse anti-Arp3 (Sigma, A5979), rabbit anti-ArpC2 (Millipore, 07227), rabbit anti-WAVE1, rabbit anti-WAVE2, guinea pig anti-N-WASP, rabbit anti-JMY, rabbit anti-WASH [32], rabbit anti-phosphotyrosine P-Tyr1000, rabbit anti-Src-P-Tyr416, rabbit anti-FAK, and mouse anti-vinculin (Sigma, V9131). Secondary antibodies were conjugated to AlexaFluor 488, 555, or 647 (Invitrogen). DNA was stained using 1μg/mL DAPI (Invitrogen), and F-actin was visualized using 0.2U/mL of AlexaFluor 488, 568, or 647-conjugated phalloidin (Invitrogen). All samples were mounted using ProLong Gold antifade (Invitrogen). Most images were captured on a Nikon Eclipse T*i* inverted microscope with Plan Apo 60X/1.40, Plan Fluor 20X/0.5, or Plan Fluor 10X/0.3 numerical aperture objectives using an Andor Clara-E camera and a computer running NIS Elements Software. Live cell imaging was performed in a 37˚C chamber (Okolab) and images collected at intervals listed in the Figure Legends (e.g., 1min to measure protrusion rates, 5min to measure motility rates). With the 60x objective, all cells were viewed in multiple focal planes, and even when Z-series were captured (at 0.2–0.4μm steps), each of the images presented in the Figures represent a single slice. Confocal images were collected using a Nikon A1R spectral confocal microscope with a PlanApo 60X/1.40 numerical aperture objective, continuously adjustable galvano scanner, and a 32-channel filter-free spectral detector. XY, XZ, and YZ projections are in depicted in several Figure panels. All cell images were processed and/ or analyzed using ImageJ software.

## Cellular F-actin quantification and line-scan analyses

Cytoplasmic F-actin levels were measured by outlining cell perimeters and calculating mean pixel intensities of phalloidin fluorescence using ImageJ. F-actin pixel intensities in cells expressing each GFP-fusion were normalized (by ratio, not subtraction) to the intensities in cells expressing GFP, which were set to an average of 1. Analyses of F-actin, fusion protein, and other antibody staining intensities at cell peripheries were performed in ImageJ using the line-scan function. 10–15μm lines were drawn perpendicular or parallel to the cell edge at different locations for multiple cells, as described in the Figure Legends. F-actin pixel intensities were normalized to the background beneath the nucleus, and the area beyond the edge of the cell was subtracted to reach 0.

## Spreading, macropinocytosis, endocytosis, and pharmacological assays

For cell spreading assays, NIH3T3 cells encoding either LAP- or mCherry-fusion proteins were grown in 6-well plates in complete media supplemented with 10mM sodium butyrate for

24h, trypsinized, and re-seeded onto glass coverslips 3h prior to fixation in 3.7% paraformalde-hyde. For EGF-induced protrusion assays, cells were cultured on coverslips in media supple-mented with sodium butyrate for 24h, transferred to serum-free DMEM for 6h, and then stimulated with 100ng/ml EGF (Sigma) for 0-5min prior to fixation. For dextran internaliza-tion assays, cells were cultured on coverslips in media supplemented with sodium butyrate for 24h, serum starved for 1h, and then incubated with 0.1mg/mL TMR-dextran 70,000MW (Invi-trogen) for 0-1h at 37˚C. Cells were rinsed three times with cold PBS before fixation. TMR-dextran intensities were quantified by outlining cell perimeters, defined by phalloidin staining, and measuring mean TMR fluorescence intensities using ImageJ. For transferrin uptake assays, cells were cultured overnight on coverslips, transfected, and incubated in growth media overnight. Cells were then serum starved for 5h, chilled on ice at 4˚C for 30min, incubated with 100ng/ml AlexaFluor568-transferrin (Invitrogen) for 45min, and shifted to 37˚C for 0–5 min to promote internalization. Cells were fixed, stained, and visualized as described above. For pharmacological treatments, cells were treated with CK666 (Millipore 182515) to inhibit the Arp2/3 complex, EHop-016 (Selleckchem s7319) to inhibit Rac activity, or with PP2 (Sell-eckchem s7008) or Saracatanib (Selleckchem s1006) to inhibit SFKs. Cells were then fixed in 2.5% paraformaldehyde, stained, and visualized as described above. For quantification, slides were coded and scored in a blinded manner.

## Cell migration assays

For wound healing assays, NIH3T3 or B16-F1 cells were cultured in 6-well plates and treated with siRNAs as discussed above. After overnight siRNA treatment, cells were seeded into spe-cialized wells that were separated by silicone spacers (Ibidi, 2-well inserts in 35mm μ-dishes) for imaging cell migration, and also into new 6-well plates for collecting cell extracts. LAP and LAP-WHIMP expression in NIH3T3 cells was induced using media containing 10mM sodium butyrate. After 24h, the silicone spacer was removed, NIH3T3 or B16-F1 cells were washed twice with warm growth media and imaged using the 10x objective at 2–3 locations at 0h, 6h, 12h, and/or 24h, while subsets of B16-F1 cells were kept in the microscope enclosure at 37˚C continuously and imaged at 5min intervals for 12h. Quantification of cell-free areas was per-formed using ImageJ. Cells cultured in parallel in 6-well plates were processed for immuno-blotting as described above.

## Protrusion and motility quantification

For scoring the efficiency of ruffling in fixed NIH3T3 or B16-F1 cells, protrusive regions or ruffles were identified by intense F-actin staining at cell edges. Cells with and without ruffles were counted using the ImageJ cell counter plugin. For measuring protrusion width, ruffles were identified in fixed cells as described above, and by phase-contrast imaging in live cells. Using the ImageJ freehand tool, a line was drawn along the protrusive area to measure widths. For quantifying protrusion speeds, kymographs made in ImageJ were used to measure the dis-tance traveled by the largest continuous protrusion within a 10min window of imaging. That value was divided by the time of that protrusive event. For monitoring cell directionality and speed, the ImageJ manual tracking plugin was used to generate individual cell tracks within wound areas. Quantification of directionality and speed was then performed using R software.

## Reproducibility and statistics

All conclusions were based on observations made from at least 3 separate experiments, while quantifications were based on data from 2–6 representative experiments. Statistical analyses were performed using GraphPad Prism software as noted in the Figure Legends. Statistics on

data sets with 3 or more conditions were performed using ANOVAs followed by Tukey's post-hoc test unless otherwise indicated. P-values for data sets including 2 conditions were determined using unpaired t-tests unless otherwise noted. Analyses of data sets involving +/- scoring used Fisher's exact test. P-values <0.05 were considered statistically significant.

## Supporting information

**S1 Fig. WHIMP orthologs are present in multiple mammals. (A)** A consensus sequence illustration of WHIMP orthologs aligned using Geneious software is shown. Non-conserved regions are depicted in white, while partially-conserved and identical residues are shown in gray and black. For genomic context, in *Mus musculus* the *Whimp* (*Gm732*) gene is found at X D, while the *Was* gene is found at X A1.1. **(B)** A phylogenetic tree of WHIMP was constructed using estimates of maximum likelihood phylogenies (PhyML) from alignments of amino acid sequences generated using Geneious software. UniProt ID numbers are listed next to the common animal names. The scale bar indicates the branch length and amino acid substitutions per site. **(C)** Primary domain organizations for mouse (*M. musculus*), rat, and Chinese hamster WHIMP, along with their sequence identities and similarities from EMBOSS Needle are shown. **(D)** The HHpred algorithm predicts sequence similarity between amino acids 154–192 of the WAVE1 α6 helix and residues 35–72 of WHIMP. H/h = helix and C/c = coil, where uppercase letters represent higher confidence predictions. **(E)** Multiple sequence alignment of the α6 WAVE homology region of WHIMP from mouse, rat, and hamster is shown.
(TIFF)

**S2 Fig. WHIMP is a weak actin nucleation-promoting factor *in vitro*. (A)** Purified maltose binding protein (MBP) tagged full-length WHIMP was subjected to SDS-PAGE followed by staining with Coomassie blue or immunoblotting with anti-WHIMP or anti-MBP antibodies. **(B)** Representative actin polymerization assays using 2μM actin, 20nM Arp2/3 complex, and the indicated concentrations of MBP, MBP-WHIMP, or MBP-WHAMM are shown. AU, arbitrary units. **(C)** Maximum actin polymerization rates were calculated from curves shown in panel B. **(D)** Times to half-maximal polymer were calculated from curves shown in panel B. **(E)** Actin polymerization assays using 2μM actin, 20nM Arp2/3 complex, and a combination of MBP, MBP-WHIMP(WCA), and MBP-WAVE2(WCA) are shown. AU, arbitrary units.
(TIFF)

**S3 Fig. WHIMP induces cytoplasmic actin assembly and localizes to membrane protrusions in B16-F1 cells. (A)** B16-F1 cells transiently transfected with plasmids encoding GFP or GFP-tagged N-WASP(WWCA), WHIMP(WCA), or full-length WHIMP were lysed and subjected to SDS-PAGE and immunoblotting with antibodies to GFP and GAPDH. **(B)** Cells treated as in A were fixed and stained with phalloidin to visualize F-actin (magenta) and DAPI to label DNA (blue). Arrows highlight GFP-WCA-expressing cells with increased cytoplasmic F-actin content, while arrowheads indicate prominent F-actin-rich protrusions in GFP-WHIMP-expressing cells. Scale bar, 20μm.
(TIFF)

**S4 Fig. NIH3T3 cell lines engineered to express fluorescently-tagged WHIMP show WHIMP localization to membrane protrusions. (A)** Diagrams of the fusion proteins used to generate fluorescent NIH3T3 cell lines are shown. The LAP-tag includes GFP and an S-peptide. **(B)** NIH3T3 cells stably encoding LAP-WHIMP or mCherry-WHIMP were treated with 10mM sodium butyrate (NaBu) for 16-18h and sorted into GFP- or mCherry-expressing populations using flow cytometry. Following outgrowth, cells were left untreated (–) or were

treated (+) with NaBu, lysed, and subjected to SDS-PAGE and immunoblotting with antibodies to WHIMP, GFP, mCherry, or Tubulin. The tagged and endogenous versions of WHIMP are indicated. **(C)** Cells treated as in B were fixed and stained with phalloidin to visualize F-actin and DAPI to label DNA. Scale bar, 100μm. **(D)** Cells induced to express LAP-WHIMP were immunoblotted with antibodies to WHIMP. The mean level of LAP-WHIMP relative to endogenous WHIMP was determined following densitometry of 6 representative blots +/-SD. **(E)** Cells induced to express LAP-WHIMP were treated with either a control siRNA or a pool of WHIMP-specific siRNAs for 48h, lysed, and subjected to SDS-PAGE and immunoblotting for WHIMP, GFP, and Tubulin. **(F)** Cells treated as in E were fixed and stained with phalloidin to visualize F-actin and with DAPI to label DNA. **(G)** Cells stably encoding LAP or LAP-WHIMP were induced, serum-starved, and stimulated with EGF for 5min. Cells were then fixed and stained with phalloidin and DAPI. Magnifications highlight LAP-WHIMP-specific enrichment at the cell periphery and co-localization with F-actin. Scale bar, 20μm. **(H)** Cells stably encoding LAP, LAP-WHIMP, or LAP-WHIMP(ΔWCA) were induced and examined live for GFP fluorescence and by phase-contrast microscopy. White lines indicate positions in which kymograph analyses were performed in Fig 5I. Scale bar, 20μm.
(TIFF)

**S5 Fig. WHIMP-associated membrane protrusions contain WAVE2 and N-WASP, but not JMY or WASH. (A)** NIH3T3 cell lines expressing a LAP tag or LAP-WHIMP were fixed and stained with antibodies to ArpC2, WAVE2, N-WASP, JMY, or WASH (red), phalloidin to visualize F-actin (magenta), and DAPI to label DNA (blue). Magnifications (right column) highlight LAP-WHIMP-specific localization to ruffles and overlap with WAVE2 and N-WASP. **(B)** LAP-expressing cells stained as in A were subjected to confocal microscopy for comparison to Fig 4D. Maximum intensity projections are shown in the large panel, and adjacent orthogonal views of YZ and XZ planes are indicated by the two gray lines. The plot profiles depict pixel intensities of LAP, Arp3, and F-actin along a 10μm line drawn near the intersection of the two gray lines. Scale bars, 20μm.
(TIFF)

**S6 Fig. WHIMP-associated membrane ruffling, tyrosine phosphorylation, and cell motility phenotypes are distinct from those associated with N-WASP or Cortactin. (A)** NIH3T3 cells stably encoding GFP, LAP-WHIMP, LAP-WHIMP(ΔWCA), GFP-N-WASP, or GFP-Cortactin were treated with sodium butyrate for 16-18h, fixed, and stained with antibodies to detect phosphotyrosines (red), phalloidin to visualize F-actin (magenta), and DAPI to label DNA (blue). White arrowheads highlight large WHIMP- and F-actin-associated membrane ruffles, while grey arrowheads point to occasional small protrusions in N-WASP- or Cortactin-overexpressing cells. Scale bar, 20μm. **(B)** B16-F1 cells were treated with siRNAs, grown in cell reservoirs separated by a 0.5mm barrier, and analyzed by phase-contrast microscopy at 0h, 6h, and 12h after removing the barrier. Control cells are shown in the top chamber, and nucleation factor-depleted cells are shown in the bottom. The white dashed box encompasses the wound area at t = 0, and the black dashed line indicates the midpoint of the cell-free area. **(C)** A directionality index was calculated as the ratio of the distance between the starting and ending point "*d*" and the trajectory "*D*" based on the cell tracks described in Fig 8E. The bars represent the mean directionality +/-SD of cells (n≥20) for each siRNA treatment. **p<0.01 (two-tailed t-test). **(D)** Cells induced to express GFP, GFP-Cortactin, LAP-WHIMP, or GFP-N-WASP were subjected to SDS-PAGE and immunoblotting with antibodies to phosphotyrosine, active Src P-Tyr416 (P-Src), and actin.
(TIFF)

**S7 Fig. WHIMP-induced membrane ruffling is partially dependent on Rac/WAVE activities. (A)** NIH3T3 cells stably-encoding LAP-WHIMP were treated with a control siRNA, independent siRNAs to WHIMP, or an siRNA to Brk1, induced to express the fusion protein, lysed, subjected to SDS-PAGE, and immunoblotted with antibodies to WAVE1, WAVE2, WHIMP, and GAPDH. The tagged and endogenous versions of WHIMP are indicated. ns, non-specific. **(B)** LAP-WHIMP cells were treated with siRNAs, fixed, and stained with phalloidin (magenta), and DAPI (blue). Scale bar, 100μm. **(C)** The % of cells with ruffles was quantified. Each bar represents the mean % of cells (n = 200–300 per condition) with ruffles from 2–3 experiments +/-SD. *** p < 0.001, * p < 0.05 (ANOVA with Tukey post-test). Note that the baseline ruffling frequency in LAP control cells is approximately 10%, irrespective of siRNA-mediated targeting of endogenous WHIMP or Brk1. **(D)** LAP-WHIMP-expressing cells were treated with 4μM EHop-016 for 5h, fixed, and stained with an antibody to WAVE1 (red), phalloidin (magenta), and DAPI (blue). The arrowhead highlights the position of a prominent WHIMP-associated ruffle. Scale bar, 20μm. **(E)** Each bar represents the % of cells (n = 105–150 per condition) with ruffles from experiments performed in part D. ** p<0.01 (Fisher's exact test).
(TIFF)

**S8 Fig. Focal adhesion proteins are recruited to WHIMP-induced Src-dependent ruffles. (A)** NIH3T3 cell lines expressing LAP, LAP-WHIMP, or LAP-WHIMP(ΔWCA), or transiently transfected with a plasmid encoding GFP-Rac1(Q61L) were fixed and stained with antibodies to vinculin (red), phalloidin (magenta), and DAPI (blue). Insets highlight LAP-WHIMP localization at ruffles and broad enrichment of vinculin staining compared to wedge-like vinculin staining in other cells. **(B)** Line-scan plots depict the mean pixel intensity of phosphotyrosine (e.g., from Fig 9A), vinculin, and F-actin staining along 15μm lines near the edges of LAP and LAP-WHIMP cells (n = 10). The edges of the cells were set to 0μm on the X-axes. **(C)** Cell lines expressing LAP-WHIMP or LAP-WHIMP(ΔWCA) were fixed and stained with antibodies to active Src P-Tyr416 (P-Src; red), phalloidin (magenta), and DAPI (blue). **(D)** Cell lines expressing LAP, LAP-WHIMP, or LAP-WHIMP(ΔWCA) were treated with DMSO or with 20μM PP2 or Saracatinib (SRCIB) for 2h, fixed, and stained with antibodies to FAK or phosphotyrosine (red), phalloidin (magenta), and DAPI (blue). Scale bars, 20μm.
(TIFF)

**S9 Fig. Pre-cropped immunoblots.** Post-cropped regions that are shown in Figs 1–10 are outlined with red boxes.
(PDF)

**S1 Data. File containing the data underlying the summary graphs.**
(XLSX)

**S1 Table. Plasmids used in this study.**
(PDF)

**S1 Video. Timelapse phase-contrast movie of siControl (top) vs siWHIMP-a (bottom) wound closure (see Fig 8).**
(AVI)

**S2 Video. Timelapse phase-contrast movie of siControl (top) vs siWHIMP-b (bottom) wound closure (see Fig 8).**
(AVI)

**S3 Video. Timelapse phase-contrast movie of siControl (top) vs siBrk1 (bottom) wound closure (see Fig 8).**
(AVI)

## Acknowledgments

We thank Jessica Fall and Nathan Jenkins for help with DNA cloning, Sarah Grout for generating stably-transfected cell lines, Ganna Brych for help with transfections, Alicia Liu and Natali Naveh in Michael O'Neill's lab for harvesting mouse organs, Pariksheet Nanda in Leighton Core's lab for assistance using R software, Chris O'Connell from the UConn Advanced Light Microscopy Facility for assistance with confocal imaging, and Vijender Singh from the UConn Computational Biology Core for assistance with bioinformatics. We also thank Katrina Velle, Aoife Heaslip, Adam Zweifach, Dave Knecht, and Campellone Lab members for their comments on this manuscript.

## Author Contributions

**Conceptualization:** Shail Kabrawala, Margaret D. Zimmer, Kenneth G. Campellone.

**Data curation:** Shail Kabrawala, Margaret D. Zimmer, Kenneth G. Campellone.

**Formal analysis:** Shail Kabrawala, Margaret D. Zimmer, Kenneth G. Campellone.

**Funding acquisition:** Kenneth G. Campellone.

**Investigation:** Shail Kabrawala, Margaret D. Zimmer, Kenneth G. Campellone.

**Methodology:** Shail Kabrawala, Margaret D. Zimmer, Kenneth G. Campellone.

**Project administration:** Kenneth G. Campellone.

**Resources:** Kenneth G. Campellone.

**Supervision:** Kenneth G. Campellone.

**Validation:** Shail Kabrawala, Kenneth G. Campellone.

**Visualization:** Shail Kabrawala, Margaret D. Zimmer, Kenneth G. Campellone.

**Writing – original draft:** Shail Kabrawala, Kenneth G. Campellone.

**Writing – review & editing:** Shail Kabrawala, Margaret D. Zimmer, Kenneth G. Campellone.

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
