## [Decision Letter · Decision Letter 0]

23 Jan 2020

Dear Dr Campellone,

Thank you very much for submitting your Research Article entitled 'WHIMP links the actin nucleation machinery to Src-family kinase signaling during protrusion and motility' to PLOS Genetics. Your manuscript was fully evaluated at the editorial level and by independent peer reviewers. The reviewers appreciated the attention to an important topic but identified some aspects of the manuscript that should be improved.

In addition to suggested revisions to the text of the manuscript by all three reviewers, Reviewer 1 also suggests a few additional analyses that may strengthen the paper. Additionally, Reviewer 3 also makes the reasonable requests that full images of the Western Blots be included as supplemental data.

Although the reviewers did not comment on this point, we would also ask you to clarify the controls for your western blots in Figures 8c and 8e; the GAPDH bands in these two panels are very similar. If they are indeed the same image, this should be called out explicitly in the figure legend and the same done for any other reused panel items.

We ask you to modify the manuscript according to the review recommendations before we can consider your manuscript for acceptance. Your revisions should address the specific points made by each reviewer.

[LINK]

Yours sincerely,

Lillian Fritz-Laylin, PhD

Guest Editor

PLOS Genetics

Gregory Barsh

Editor-in-Chief

PLOS Genetics

**Comments to the Authors:**

Reviewer #1: Summary

The manuscript by Kabrawala et al reports a novel Nucleation Promoting Factor (NPF), which the authors have named WHIMP. The study utilizes bioinformatics, as well as biochemical and cell biological approaches to conclude that WHIMP drives membrane protrusion and cell migration by activating Src family kinases. The authors also conclude that a complex interplay exists between NPFs, as disruption of the WAVE complex impacts WHIMP-dependent membrane ruffling.

The authors have conducted a robust, thorough study (as evidenced by the body of supplemental data and nine main figures). The finding that WHIMP is a bona fide NPF is significant enough in my mind to warrant interest from the field. Though WHIMP is absent from the human genome, novel NPFs may yet be discovered in light of the present findings. Additionally, mouse cell lines like NIH 3T3 and B16-F1 are used routinely to model cellular behavior, and as such it is critical to understand the complete complement of cytoskeletal regulators in these cells. The manuscript is generally well written and most of the authors’ conclusions are well supported by the data. I support publication of this manuscript, but would like to hear the authors’ views on the following points and to provide a few pieces of key additional data, where appropriate.

Major points

It is clear that the authors have responded to a number of critiques regarding this manuscript. The major outstanding concern that I have is that many of the conclusions are based on ectopic overexpression of WHIMP. It is clear that transfected cells demonstrate a number of robust phenotypes consistent with WHIMP’s ability to promote dendritic actin nucleation at membranes, but the contribution of endogenous WHIMP to membrane protrusion and F-actin generation remains unclear.

1) Much of the paper is dedicated to understanding how GFP-WHIMP(WCA), mCherry-WHIMP, or LAP-WHIMP affect cellular protrusion and F-actin generation. Have the authors considered whether the overexpression of WHIMP above physiological levels might be inducing it to mis-localize to protrusions or become hyper-activated, leading to aberrant dorsal ruffle production? Are the authors able to stain endogenous WHIMP with their antibody? Figure 5 demonstrates that NIH3T3 fibroblasts expressing endogenous levels of WHIMP don’t ruffle nearly as much as when LAP-WHIMP has been expressed on top of endogenous WHIMP. A knockdown/rescue approach might work to address this point, as the authors could sort siRNA WHIMP cells re-expressing mCherry-WHIMP based on fluorescence intensity. The authors can identify a population of cells that expresses mCherry-WHIMP near endogenous levels. They can then better assess the localization of mCherry-WHIMP and its ability to induce dorsal ruffling and membrane protrusion relative to control and siWHIMP cells. This is a major point that in my mind warrants additional experimentation.

One brief note on Supplementary Figure 8: This data seems supportive of the authors’ overall conclusions. siRNA knockdown of WHIMP decreases ruffling. However, the siCtl cells are overexpressing LAP-WHIMP. The authors should also include siCtl NIH3T3 cells expressing endogenous levels of WHIMP as well as siWHIMP cells. It may be that the percentage of cells with ruffles is not so different in siCtl NIH3T3 cells and WHIMP KD NIH3T3 cells.

2) The authors should consider more thorough analysis of their siRNA WHIMP cells compared to siCtl NIH3T3 expressing endogenous WHIMP. Are membrane protrusions and/or dynamics, or dorsal ruffles compromised? Are F-actin levels altered? What about p-SFK and p-Tyr levels? The siRNA data in Figures 6 and 7 are quite nice and ultimately suggest that these additional investigations will likely be fruitful. The experiments proposed above may yield additional mechanistic insight into how WHIMP alters cell motility and cellular architecture. It appears the authors have conducted some of these studies with B16-F1 cells in Figure 7 and 8F, which supports their overall interpretation of WHIMP function. Conducting similar studies in NIH3T3 cells should therefore be fairly straightforward. It may also be worthwhile to report the relative difference in endogenous WHIMP expression in NIH3T3 versus B16-F1 cells at baseline. This may be interesting given that B16-F1 cells seem more motile and ruffle more than NIH3T3 cells at baseline (e.g. Figure 5B versus Figure 7G and Figure 6E versus Figures 7 C-D).

Minor points

1) The authors should consider editing their schematic diagram in Figure 9E. It is not clear at first glance what the blue arrows indicate to the reader.

2) It is sometimes unclear why the authors switch between NIH3T3 and B16-F1 cells. For example, Figure 8 is largely conducted with NIH3T3 cells, aside from panel F. Should the data be re-organized? Alternatively, a short rationale would be useful to the reader.

3) The authors conclude the abstract by making a statement about ‘feedback control’ of WASP-family proteins/SFKs. While their data is supportive of this idea, especially with respect to the p-Tyr signaling, one can imagine alternative possibilities. For Figure 7 (especially panel G): Perhaps WHIMP and WAVE1/2 are both required in parallel to initiate the dramatic ruffle-based protrusion. Figure 9C indicates that LAP-WHIMP cells still generate a significant seeming ruffle phenotype upon SFK inhibition. While SFK may activate (and be activated by) WHIMP, this seems to imply that WHIMP’s activity is to some extent SFK-independent. Perhaps SFK can activate WHIMP and WAVE in parallel, but WHIMP is required to sustain SFK activity, which then acts as a positive feedback mechanism to sustain WAVE. A discussion of these points (and alternative organizations of the pathway) may fit well at the end of the paper (and inform the construction of Figure 9E, as mentioned above).

4) Overall the discussion section of the paper is well written. However, it reads like a lengthy recap of the results with a first paragraph that largely touches on already introduced information from the introduction. The authors may want to consider streamlining this section a bit more so that they can unpack the concept of NPF coordination and how WHIMP might intersect with GTPase/SFK/WAVE signaling, especially given that there are many ways in which such regulation might work. I realize that much of this comes down to personal preference and style, so I would not consider this a requirement for publication.

Reviewer #2: In the manuscript entitled “WHIMP links the actin nucleation machinery to Src-family kinase signaling during protrusion and motility” Kabrawala et al. discover a new actin nucleation promoting factor encoded on the X-chromosome, WHIMP. WHIMP is considered a new WASP family member and using bulk assays the authors show that it promotes actin assembly by activating the Arp2/3 complex in fluorescence-based bulk actin assembly assays. Using several mammalian cell lines the authors show that the presence of more WHIMP in cells promotes actin assembly behaviors mediated by the Arp2/3 complex (i.e. cell ruffling, migration, wound healing), and that less WHIMP protein results in reduced motility and protrusion. Using pharmacological agents, the authors show that WHIMP also activates cellular tyrosine kinase signaling cascades. This work addresses all the experiments this reviewer can think of as required to characterize a new Arp2/3 complex nucleation promotion factor as well as probes intriguing signaling / cell regulatory mechanisms. Congratulations to the authors to this work. Below are specific concerns and points of clarification:

• Overall the introduction reads long, but there is value in explaining all the caveats of the WASP family proteins.

• Referring to WASP and Cortactin as “nucleation factors” is confusing. Neither is a bona fide actin nucleation protein (i.e. the Arp2/3 complex, formins, etc), but rather “nucleation promoting factors” which indirectly stimulate actin assembly by activating the canonical nucleation protein.

• A nice addition/point to make around line 64/65 could be mentioning the balance between Arp2/3 and formin (mDia2) activities by Dip/Wish/Spin90.

• The authors should consider reordering the sequences found in Figures 1A and 1B. For reader clarity, it might be easier to make the comparison of WHIMP to WASP if WHIMP was on the first line rather than the last line. Or possibly reorder to make the WHIMP sequence closer to WAVE1 since the sequence homology is highest on the amino acid level?

• The data presented in Figure 2B imply that WHIMP cannot nucleate actin assembly on its own. I am confused by wording of other information in Figure 2S which implies WHIMP can nucleate actin alone, but only at high concentrations. Noting the concentration of WHIMP that is expected to be closest to the physiological concentration might clarify this finding for readers.

• The authors should clarify further whether WHIMP is actually capable of binding profilin-actin or if this is an assumption due to the lack of PLP peptides (lines 133-134).

• Can the authors please comment on whether the reduced protein and gene expression of WHIMP in Figures 1D and 1E is because it is on the X chromosome? The N2A and HT22 cells used are derived from male mice, so they effectively have one less dose of X-chromosome? Perhaps information can be determined or tracked from the population of mice used in 1D?

• How many cells did GFP-WHIMP localize to cell peripheral membranes (lines 216-217). Was very few or were they easy to find?

• Figure 3C the significant increase in actin intensity with GFP-WHIMP constructs doesn’t actually look like it is 25%... are these analyses done in a blind manner? The authors should clarify and also add some text about their positive control (N-WASP).

• There is a significant amount of text devoted to the WHIMP micropinocytosis assay, however all this data is placed in the supplemental file. Either this text should be reduced/bottom-lined or Figure S7 should be introduced into the main text. The finding/dissection of the role of WHIMP in nonspecific micropinocytosis v. RME is valuable and well-done. So perhaps the latter.

• As in the introduction, the wording of nucleation factor should be changed to nucleation promoting factors for clarity.

Reviewer #3: This is a well-executed study, showing the discovery and characterisation of a new WASP-family protein that is specific to a subset of mammals. The discovery of WHIMP is interesting to the field and has been well documented and presented here. The authors demonstrate that WHIMP is widely expressed in mouse tissues and cells and that it contributes to actin nucleation at the leading edge of cells. They link WHIMP also to dorsal ruffles, which contrubute to macropinocytosis and to src activation. This is a very intriguing new member of the WASP-family and the characterisation provided by this study is convincing and clear. I have only minor comments.

1. Can the authors make it more clear in their abstract which organisms express WHIMP? e.g. is it a subset of mammals?

2. Is WHIMP near to WASP on the X chromosome? Did it likely arise from a gene duplication of WASP?

3. Can the authors provide scans of original western blots (if they haven't already) as supplementary information.

4. Do the authors have the possibility to do quantitative RT-PCR to provide any insight into how abundant WHIPM is? The provide a yes/no gel image, which is OK, but if they can do Q-PCR it would be better.

**Have all data underlying the figures and results presented in the manuscript been provided?**

Reviewer #1: Yes

Reviewer #2: Yes

Reviewer #3: No: I am not sure whether this is provided- I only had access to the main manuscript and supplementary figures- but I think that they should provide the original scans of the western blots at least.

PLOS authors have the option to publish the peer review history of their article (what does this mean?). If published, this will include your full peer review and any attached files.

Reviewer #1: No

Reviewer #2: No

Reviewer #3: No

---

## [Editor Report · Decision Letter 1]

22 Feb 2020

Dear Dr Campellone,

We are pleased to inform you that your manuscript entitled "WHIMP links the actin nucleation machinery to Src-family kinase signaling during protrusion and motility" has been editorially accepted for publication in PLOS Genetics. Congratulations!

Yours sincerely,

Lillian Fritz-Laylin, PhD

Guest Editor

PLOS Genetics

Gregory Barsh

Editor-in-Chief

PLOS Genetics

Comments from the reviewers (if applicable):

The added text to legend for Figure 9 (Figure 8 in the original submission) reads as follows: "The samples in E are the same as those used in the middle panel of E”. This needs to be corrected before publication.

**Data Deposition**

http://datadryad.org/submit?journalID=pgenetics&manu=PGENETICS-D-19-01977R1

**Press Queries**

---

## [Editor Report · Acceptance letter]

12 Mar 2020

PGENETICS-D-19-01977R1 

WHIMP links the actin nucleation machinery to Src-family kinase signaling during protrusion and motility 

Dear Dr Campellone, 

We are pleased to inform you that your manuscript entitled "WHIMP links the actin nucleation machinery to Src-family kinase signaling during protrusion and motility" has been formally accepted for publication in PLOS Genetics! Your manuscript is now with our production department and you will be notified of the publication date in due course.

With kind regards,

Jason Norris

PLOS Genetics

On behalf of:
